# FROM COMPRESSION TO EXPRESSION:
# A LAYERWISE ANALYSIS OF IN-CONTEXT LEARNING

## ABSTRACT

In-context learning (ICL) enables large language models (LLMs) to adapt to new tasks without weight updates by learning from demonstration sequences. While ICL shows strong empirical performance, its internal representational mechanisms are not yet well understood. In this work, we conduct a statistical geometric analysis of ICL representations to investigate how task-specific information is captured across layers. Our analysis reveals an intriguing phenomenon, which we term *Layerwise Compression-Expression*: early layers progressively produce compact and discriminative representations that encode task information from the input demonstrations, while later layers express these representations to incorporate the query and generate the prediction. This phenomenon is observed consistently across diverse tasks and a range of contemporary LLM architectures. We demonstrate that it has important implications for ICL performance—improving with model size and the number of demonstrations—and for robustness in the presence of noisy examples. To further understand the effect of the compact task representation, we propose a bias-variance decomposition and provide a theoretical analysis showing how attention mechanisms contribute to reducing both variance and bias, thereby enhancing performance as the number of demonstrations increases. Our findings reveal an intriguing layerwise dynamic in ICL, highlight how structured representations emerge within LLMs, and showcase that analyzing internal representations can facilitate a deeper understanding of model behavior.

## 1 INTRODUCTION

In-context learning (ICL) (Brown et al., 2020; Dong et al., 2022) has emerged as a powerful capability of large language models(LLMs), allowing them to perform new tasks by conditioning on a few input-output examples without weight updates. Despite being trained solely for next-token prediction, LLMs exhibit strong empirical performance across a wide range of NLP tasks through this mechanism. For example, pretrained LLMs can make correct predictions based on a sequence of input-separation-output pairs that encode semantic mappings. Given the same query token, the model can make different predictions based on the task defined by the demonstrations, such as $d$ and $C$ for the following two tasks,

$$\underbrace{(a \rightarrow b,\ b \rightarrow c}_{\text{Next Letter}},\ c \rightarrow?), \quad \underbrace{(a \rightarrow A,\ b \rightarrow B}_{\text{To Uppercase}},\ c \rightarrow?) \tag{1}$$

Recent research has advanced several theoretical perspectives on explaining why ICL works—viewing ICL as posterior inference (Xie et al., 2021), implicit meta-learning (Chen et al., 2021), internal optimization (Von Oswald et al., 2023) or mechanistic interpretations(Olsson et al., 2022). However, the underlying mechanism of how LLMs distinguish different tasks and use this information to guide their output remains unclear for ICL. To address this gap, we focus on the hierarchical feature learning across layers and formulate the following research question:

*How do LLMs extract and differentiate task information from shallow to deep layers during in-context learning?*

To investigate how task-specific information evolves across layers, we conduct a statistical geometric analysis of ICL representations across multiple tasks. Specifically, we consider a set of $T$ ICL tasks,

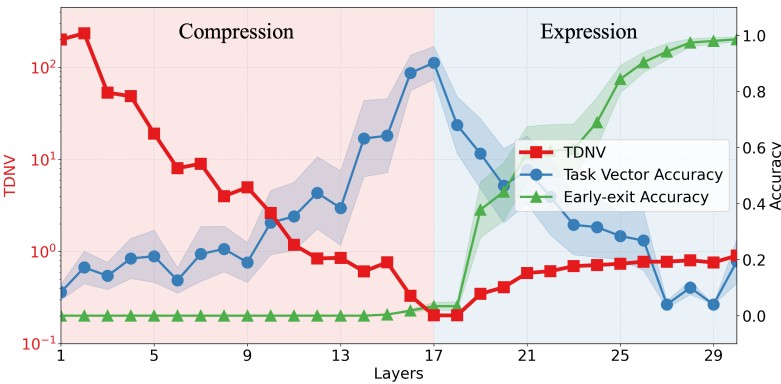

Figure 1: **Layer-wise compression to expression** in ICL representations. TDNV first decreases then increases from shallow to deep layers, splitting the model into compression and expression stages. During the compression stage, task vector accuracy increases as task information is progressively extracted from demonstration pairs. During the expression stage, early-exit accuracy increases as output information is progressively decoded based on the input query. Refer to Appendix G for detailed explanation of task vector and early-exit accuracy.

each associated with a distribution $\mathcal{D}_t$ for input-output pairs. Then for each task $t$, we randomly sample $K$ input-output pairs (also referred to as *demonstrations*) from $\mathcal{D}_t$, which are combined with a query to form an ICL *instance*. We construct multiple such instances per task following this procedure. To quantify how the model compresses task information in its internal representations, we examine two key aspects: (1) how instances from the same task are clustered together, and (2) how instances from different tasks are distinguished from each other. Our analysis reveals an intriguing phenomenon, which we term the **Layerwise Compression-Expression**, summarized as:

---

**Layerwise Compression-Expression Phenomenon**

*LLMs exhibiting ICL capabilities organize their layers into two parts with distinct behaviors: a compression part and an expression part. The early layers, comprising the compression part, progressively produce compact and discriminative representations that capture task information from the input demonstrations. The later layers, forming the expression part, apply these compact representations to the query to generate the output.*

---

Specifically, we introduce a metric called Task-Distance Normalized Variance (TDNV)[1] that measures the ratio of within-task variance to between-task distance: within-task variance indicates how well the representation from the same task are compressed, while between-task distance reflects the separation from different tasks. A lower TDNV indicates that representations of the same task samples are similar and representations of different task samples are distinguishable. Thus, TDNV serves as an effective method of how well the task information is compressed. By measuring TDNV across transformer layers, we can track how the model progressively encodes and transforms task information throughout its architecture.

As shown in Figure 1, TDNV first decreases and then increases from shallow to deep layers, splitting the model into compression and expression stages. To further support our hypothesis, we conduct causal interventions and measure task vector accuracy (Hendel et al., 2023) and early-exit accuracy (Xin et al., 2020; Jiang et al., 2024b) across layers to track task-specific and output-specific information. Task vector accuracy measures zero-shot ICL performance when injecting intermediate layer hidden states extracted under ICL settings. Early-exit accuracy measures performance by directly applying the final-layer classifier to intermediate hidden states. During compression, task vector accuracy increases while early-exit accuracy remains low, indicating that the model is compressing representations to encode task information. During expression, task vector accuracy decreases while early-exit accuracy rapidly increases, indicating that the model begins to incorporate query information and decode output-specific representations. As we show, this behavior has important implications for ICL performance and robustness.

---

[1]Following a similar conceptual framework to Class-Distance Normalized Variance (CDNV) (Galanti et al., 2021) by viewing each task as a class.

**Contributions.** Our main contributions can be summarized as follows:

- By analyzing the hidden representations of ICL, we conceptualize and extensively examine the Layerwise Compression-Expression phenomenon. Our results show that it is prevalent across model architectures (transformer and state-space models) and task domains (symbolic, language understanding and multimodality), and emerges during the training process.
- We show that larger models and more demonstrations lead to more compressed task representations, explaining why larger models and longer contexts yield better performance. To further understand the compressed representation, we propose a bias-variance decomposition and provide a theoretical analysis showing how attention mechanisms contribute to reducing both variance and bias, thereby enhancing performance as the number of demonstrations increases.
- We show that noisy demonstrations result in less compressed representations and a corresponding drop in performance. However, the representations remain distinguishable with a certain amount of noise, which helps explain the robustness of ICL. Moreover, we find that errors in early demonstrations can be suppressed by later examples, and that errors in later demonstrations lead to less compressed representations than those in early ones. This highlights the recency effect (Kossen et al., 2024; Yu & Ananiadou, 2024) and the key role of later demonstrations.
- Motivated by our analysis, we propose task-vector contrastive fine-tuning method to further compress task vectors and reduce TDNV. Fine-tuning GPT-2 models on symbolic ICL tasks with this approach yields 20% improvement on average in task-vector accuracy over standard finetuning.

**Significance of the Finding.** Our analysis provides a new perspective on why *decoder-only* LLMs trained for next-token prediction can serve as flexible architectures for a wide range of tasks. Despite lacking an explicit bottleneck layer, these models exhibit behavior reminiscent of *encoder-decoder* architectures: early layers distill task information from demonstrations into compact representations, while later layers decode these representations into query-specific outputs. The compression stage aligns with the Information Bottleneck (IB) principle (Saxe et al., 2019; Kawaguchi et al., 2023), which posits that efficient neural representations are achieved by compressing inputs to retain only task-relevant information while discarding redundant or irrelevant details. However, standard IB theory focuses exclusively on the compression phase and is primarily developed in the context of classification problems. Our work also provides justification for previous pruning studies (Men et al., 2024; Luo et al., 2025), which show that deeper layers tend to be more redundant and can be safely skipped, whereas skipping earlier layers often results in significant performance degradation.

## 2    RELATED WORKS

**Layerwise Representations.** Prior works (Ben-Shaul & Dekel, 2022; Fang et al., 2021; Wang et al., 2023b; Rangamani et al., 2023; He & Su, 2024; Zhou et al., 2025) investigated the role of different layers in feature learning. They revealed that in classification task, intermediate layer features become increasingly linearly separable and exhibit Neural Collapse ($\mathcal{NC}$), indicating **monotonic** feature compression with depth. In contrast, we hypothesize that decoder-only ICL models follow a **dual** process where shallow layers compress information and deeper layers re-express it, and intermediate layers achieve maximal compression.

**In-Context Learning Interpretability.** Numerous studies have investigated the mechanisms underlying ICL (Xie et al., 2021; Chen et al., 2021; Von Oswald et al., 2023; Dai et al., 2022; Ahn et al., 2023; Olsson et al., 2022), spanning perspectives of Bayesian inference, meta-learning, and optimization. Our work instead analyzes through layer-wise representations. In addition, Doimo et al. (2024) examine the geometry of ICL representations by clustering intermediate features according to semantic subjects of input, whereas our findings differ by focusing on the underlying tasks induced by input–output pairs.

**Task Representations.** Various compact representations capture ICL tasks, including task vector (Hendel et al., 2023), function vector (Todd et al., 2023), and state vector (Li et al., 2024), which guide model behavior by injecting hidden states. Other works explore compositional and latent-space manipulation (Shao et al., 2023; Liu et al., 2023b). Prior studies focus on single-task representations, whereas we provide a layer-wise geometric analysis of representations.

A more comprehensive discussion of the related works can be found in Appendix B.

## 3 PRELIMINARIES

In this section, we first formally set up the layer-wise representations of in context learning in Section 3.1, followed by introducing the metrics for measuring within-task compression of features at each layer in Section 3.2.

### 3.1 PROBLEM SETUP

**Layerwise ICL Representations.** For ICL task, we are given $(i)$ $K$ demonstrations, denoted as $\mathcal{S}_K = \{s_1, s_2, \ldots, s_K\}$, where each demonstration $s_k = (x_k \to y_k)$ consists of an input token $x_k$, a separator token "$\to$", and a label token $y_k$; and $(ii)$ a query input $\mathcal{X} = (x_q \to)$. We refer to the demonstration-query pair $(\mathcal{S}_K, \mathcal{X})$ as an ICL *instance*. An LLM $f$ performs ICL by processing the instance $(\mathcal{S}_K, \mathcal{X})$ as a next-token prediction task. Let $\boldsymbol{Z}^{(\ell)} \in \mathbb{R}^{d \times p}$ denote the hidden representations at layer $\ell$ for the instance, where $p$ denotes the sequence length and $d$ represents the dimension of the hidden representation. The layerwise update with $f$ is performed as

$$\boldsymbol{Z}^{(\ell+1)} = f_{\theta^{(\ell)}}(\boldsymbol{Z}^{(\ell)}), \quad \text{for } \ell = 0, 1, \ldots, L-1, \tag{2}$$

where $f_{\theta^{(\ell)}} : \mathbb{R}^{d \times p} \to \mathbb{R}^{d \times p}$ denotes the transformation—such as self-attention and a multi-layer perceptron in a Transformer—within the $\ell$-th layer, parameterized by $\theta^{(\ell)}$.

For autoregressive models, the final prediction is produced by applying a classifier on the representation of the last separation token in the final layer $\boldsymbol{Z}^{(L)}$, which predicts the label $y_q$ of the query input. Since this token summarizes the entire context, we use its hidden representation as the ICL representation at layer $\ell$, denoted by $\boldsymbol{h}^{(\ell)}$ to simplify the notation. This vector is also referred to as the *task vector* in (Hendel et al., 2023). Throughout the remainder of this paper, we use $\boldsymbol{h}^{(\ell)}$ to analyze layer-wise behavior and information encoding in the ICL process.

### 3.2 METRIC FOR REPRESENTATION COMPRESSION

To analyze how models distinguish between different tasks, we consider $T$ ICL tasks, each with a task-specific data distribution $\{\mathcal{D}_t\}_{t=1}^T$. For each task $t$, we sample $N$ ICL instances of form $(\mathcal{S}_K, \mathcal{X})$ from $\mathcal{D}_t$. For each instance, we use the hidden representation of the last token at layer $\ell$ as the representation of the inputs, denoted as $\boldsymbol{h}_{i,t}^{(\ell)} \in \mathbb{R}^d$ for the $i$-th instance from task $t$.

The study of feature compression and discrimination (Yu et al., 2020; Papyan et al., 2020; Zhu et al., 2021; Zhai et al., 2020; Galanti et al., 2021; Jiang et al., 2023b) has recently gained significant attention in representation learning. Inspired by this line of work, we analyze how models compress task information in their internal representations by examining two key aspects.

- We quantify how samples from the same task cluster together by using the **within-task variance**

$$\mathrm{var}_t^{(\ell)} = \frac{1}{N} \sum_{i=1}^N \|\boldsymbol{h}_{i,t}^{(\ell)} - \overline{\boldsymbol{h}}_t^{(\ell)}\|_2^2, \quad \text{where} \quad \overline{\boldsymbol{h}}_t^{(\ell)} = \frac{1}{N} \sum_{i=1}^N \boldsymbol{h}_{i,t}^{(\ell)}. \tag{3}$$

  It measures how well representations from the same task are compressed toward their task mean. Specifically, when this value decreases, it indicates that features within each task are more tightly compressed around their respective means.

- To quantify how effectively samples from different tasks are distinguished from each other, we use the **between-task distance** of two tasks $t$ and $t'$ as $\|\overline{\boldsymbol{h}}_t^{(\ell)} - \overline{\boldsymbol{h}}_{t'}^{(\ell)}\|_2^2$. It measures the distance between the centers of different tasks and the features of each task become more separable as this distance increases.

We then use a metric inspired by the class-distance normalized variance used in classification tasks Galanti et al. (2021), referred to as Task-Distance Normalized Variance (TDNV) here to measure the ratio of within-task variance to between-task distance:

$$\mathrm{TDNV}^{(\ell)} := \sum_{t=1}^T \sum_{\substack{t'=1 \\ t' \neq t}}^T \frac{\mathrm{var}_t^{(\ell)} + \mathrm{var}_{t'}^{(\ell)}}{2\|\overline{\boldsymbol{h}}_t^{(\ell)} - \overline{\boldsymbol{h}}_{t'}^{(\ell)}\|_2^2}, \quad \forall \ell \in [L]. \tag{4}$$

The decrease of TDNV indicates more compressed and discriminated feature for the ICL task.

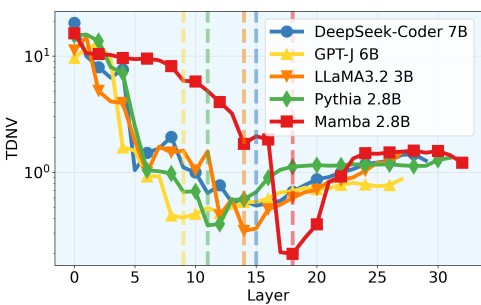 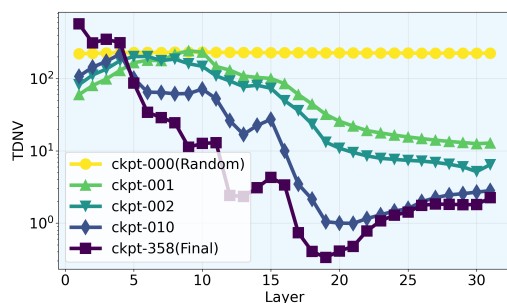

Figure 2: Layerwise TDNV of different model architectures, including decoder-only transformers and state-space models.

Figure 3: Layerwise TDNV during training process. The phenomenon emerges and intensifies with training progress.

## 4 LAYERWISE COMPRESSION-EXPRESSION DYNAMIC

In this section, we examine the dynamics of layer-wise representation under the ICL setting, a phenomenon we termed *Layerwise Compression-Expression*. The subsequent sections validate and explore this phenomenon in detail across various conditions. Specifically, Section 4.1 demonstrates that it occurs universally across different architectures and tasks. Next, we analyze key factors influencing this phenomenon, including model size (Section 4.2) and demonstration noise (Section 4.3).

### 4.1 PREVALENCE OF LAYERWISE COMPRESSION-EXPRESSION

To validate whether *Layerwise Compression-Expression* is a general mechanism of ICL, we evaluate it across different LLM model architectures and tasks. Unless otherwise specified, we use Deepseek-coder-7B (Guo et al., 2024) as our default model. For each task, we sample $N = 100$ instances, setting the default number of demonstrations to $K = 15$. We ran three independent experiments with different ICL samples, and the TDNV error bar is omitted due to negligible variance.

**Universality across model architectures.** Following Hendel et al. (2023), we first evaluate the algorithmic domain, including 5 tasks (copy letter, next letter, to uppercase, previous letter and next 2 letter). Detailed descriptions of all tasks are provided in Appendix A. As shown in Figure 2, the TDNV metric consistently exhibits a U-shaped trend—first decreasing then increasing—across two distinct architectural families: ($i$) Decoder-only transformers, including Llama3 (Grattafiori et al., 2024), Pythia (Biderman et al., 2023), GPT-J (Wang & Komatsuzaki, 2021) and Deepseek-coder (Guo et al., 2024). ($ii$) State-space models, specifically Mamba (Gu & Dao, 2023). This phenomenon holds even in the absence of attention, as evidenced by Mamba, indicating that the mechanism is not specific to the transformer architecture. Additional experiments on Gemma2 and Mistral models are included in Appendix C.

**Universality across task domains.** To evaluate the generality of the phenomenon beyond algorithmic settings, we examine three additional task categories: ($i$) Symbolic ICL. We adopt the linguistic, translation, and knowledge domains from Hendel et al. (2023), where TDNV consistently exhibits a U-shaped trend (Figure 4). ($ii$) Language Understanding ICL. Beyond only short phrases, we evaluated TDNV on a natural language dataset with longer sentences. Each sentence can be analyzed across multiple attributes: length, semantic polarity, tense, sentence type, subject person, and entity type. We adopt Llama3 8B (Grattafiori et al., 2024) to predict the attribute label (e.g., positive or negative) for a query sentence based provided demonstrations with labels of a specific attribute (e.g., semantic polarity). As shown in Figure 5, the TDNV also exhibits a U-shaped trend, with the most compressed representation shifting to a later layer (around layer 28), indicating that longer sentences require more layers for effective task compression. ($iii$) Multimodality ICL. We further extend to a vision–language setting using a 2-D shape dataset, where each image contains a shape with four attributes (color, shape, size, texture). We adopt the Qwen-VL (Bai et al., 2023) model to predict the attribute label (e.g., red or green) for a query image based provided demonstrations with labels of a specific attribute (e.g., color). As shown in Figure 6, the TDNV metric again exhibits a U-shaped curve. Across all settings, increasing the in-context length $K$ leads to more compact internal representations with lower TDNV values. Complete task specifications are provided in Appendix A.

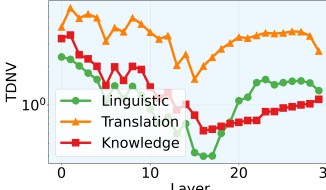
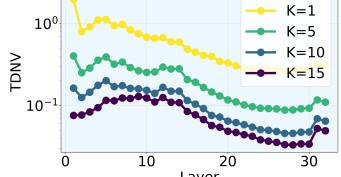
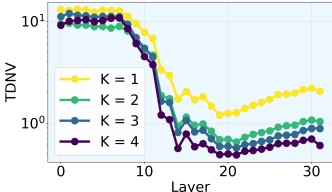

Figure 4: Symbolic ICL.    Figure 5: Language Under-    Figure 6: Multimodality ICL.
standing ICL.

**Emergence during training.** To verify that the phenomenon emerges only in trained models and not in randomly initialized ones, we evaluate on the Amber models of LLM360 family (Liu et al., 2023c), which provide intermediate checkpoints throughout the training process. As shown in Figure 3, models with random initialization exhibit flat TDNV values across all layers, indicating no structure of information compression. As training proceeds, the TDNV curve transitions into a distinct U-shape curve. This suggests the phenomenon only emerges as a result of training.

## 4.2 Scaling Up Model size Leads to More Compression

To explore how model size influences information compression, we analyze Pythia models ranging from 14M to 410M parameters in terms of both layerwise TDNV and performance (as shown in Figure 7). We evaluate ICL performance from two perspectives: (1) the regular few-shot setting, referred to as ICL, and (2) the task-vector (TV) setting—i.e., zero-shot ICL using a task vector patched from the best-performing layer $\hat{\ell}$ identified under the few-shot setting—referred to as TV ICL. Higher accuracy in either setting indicates better performance. Additionally, we report zero-shot accuracy without any task-vector information, referred to as the baseline. We find that larger models tend to produce more compressed and discriminative representations in the middle layers, indicating a stronger ability to extract task information from the demonstrations, thereby achieving better performance in both ICL and task-vector ICL.

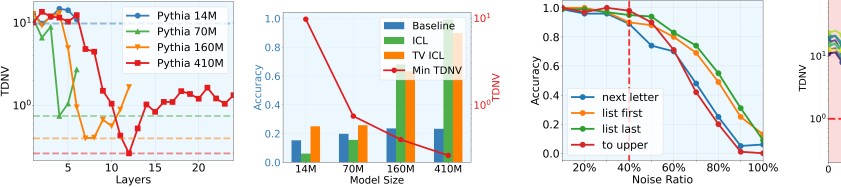

Figure 7: Effect of model size on layerwise    Figure 8: Effect of noisy demonstrations on ICL
TDNV and ICL performance.    performance and layerwise TDNV.

## 4.3 Compression-to-Expression under Noisy Demonstrations

Demonstrations in real-world scenarios are often noisy, with some input-output pairs failing to accurately represent the intended task. Despite this, ICL demonstrates notable robustness to such noisy demonstrations. As illustrated in Figure 8 left, performance remains largely unaffected even when the noise ratio reaches up to 40%, where the noise ratio is defined as the proportion of incorrect input-output pairs relative to the total number of pairs. To understand this robustness, we explore it through the lens of information compression.

We plot the layerwise TDNV under varying noise ratios in Figure 8 right and highlight two key observations: ($i$) higher noise ratios consistently lead to increased TDNV across all layers, indicating that noisy demonstrations impair the model's ability to compress and extract task-relevant information. In the extreme case of 100% noise—where inputs and labels are completely uncorrelated—the model receives no meaningful task signal, and the characteristic compression-to-expression pattern disappears across layers. ($ii$) When the noise ratio remains below 40%, the minimum TDNV values stay below 1, indicating that within-task variance is still smaller than between-task distance. This allows task representations to remain distinguishable, resulting in minimal performance degradation. This observation explains the robust performance at noise ratios below 40%. However, beyond 40% noise, task representations become increasingly entangled, causing performance to decline rapidly.

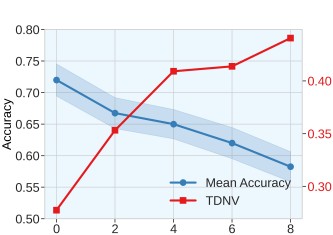 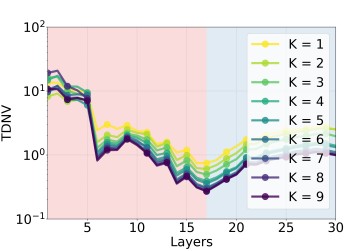 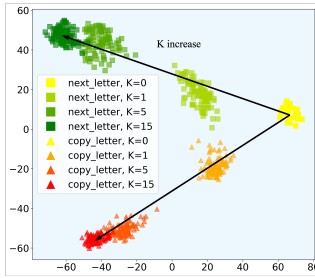

Figure 9: Layerwise TDNV and ICL accuracy under different perturbation positions.

Figure 10: Layerwise TDNV under different number of demonstrations $K$.

Figure 11: PCA of task vectors as number of demonstrations $K$ increase.

**The position of noisy demonstration affects compression.** Unlike conventional machine learning, the order of demonstrations has a significant impact on model performance in ICL (Liu et al., 2023a; Lu et al., 2021; Zhou et al., 2024). As illustrated in Figure 9, perturbing demonstrations that appear later in the sequence causes a larger performance drop and higher TDNV values. These perturbations result in less compressed task vectors, indicating that later demonstrations play a more crucial role than earlier ones in helping the model extract task information. To gain deeper insight, we perform a fine-grained analysis by computing layerwise TDNV for each separator token, referred to as grid TDNV in Appendix D.2.

## 5 BIAS-VARIANCE DECOMPOSITION OF TASK VECTORS

In this section, we analyze the effect of the number of demonstrations and study the task vectors in the middle layer—where the representation is most compact (i.e., exhibits the smallest TDNV)—using a bias–variance decomposition.

**Increasing in-context lengths lead to more compressed representations.** We first evaluate how the number of demonstrations affects the geometry of ICL representations using layerwise TDNV. The results in Figure 10 show that increasing the number of demonstrations $K$ consistently reduces TDNV across all layers. This indicates that as more demonstrations are provided, the within-task variance of task vectors decreases while the between-task distance increases. This explains why increasing the number of demonstrations improves performance—it leads to more compressed and more distinct representations in the intermediate layers.

**Bias-Variance decomposition.** In Figure 11, we present a PCA visualization of the most compressed layer for two different tasks that share the same query; see an illustrative example in (1). When no demonstrations are provided ($K = 0$), both tasks produce the same vectors that reflect the prior of the pretrained model. As $K$ increases, we observe an intriguing phenomenon: ($i$) different tasks induce task vectors in distinct directions, yet each task follows a consistent direction; ($ii$) the variance within each task decreases. Based on this observation, we decompose the task vector $\boldsymbol{h}_{i,t}(K)$ (where we highlight the dependence on the number of demonstrations $K$ and omit the superscript $(\ell)$) of each instance into the following components

$$\boldsymbol{h}_{i,t}(K) = \boldsymbol{\mu}_t(\infty) + \underbrace{\boldsymbol{\mu}_t(K) - \boldsymbol{\mu}_t(\infty)}_{\text{bias}} + \underbrace{\boldsymbol{h}_{i,t}(K) - \boldsymbol{\mu}_t(K)}_{\text{variance}}, \tag{5}$$

where $\boldsymbol{\mu}_t(K) = \mathbb{E}_i[\boldsymbol{h}_{i,t}(K)]$ denotes the mean of the task vector obtained from $K$ demonstrations, and $\boldsymbol{\mu}_t(\infty) = \lim_{K \to \infty} \mathbb{E}_i[\boldsymbol{h}_{i,t}(K)]$ represents the mean of the task vector obtained from infinitely many possible demonstrations (ignoring the practical limitations of context length in real-world LLMs), which maybe referred to as the *ideal* task vector.

**How well does the mean task vector encode task information?** Unlike the classical bias–variance decomposition—where the mean of multiple models often outperforms individual models due to the ensemble effect—the setting here is more nuanced. The mean task vector $\boldsymbol{\mu}(K)$ is averaged not only over different demonstrations but also over different queries. Therefore, it is not immediately clear whether the mean task vector still encodes useful task information—and if it

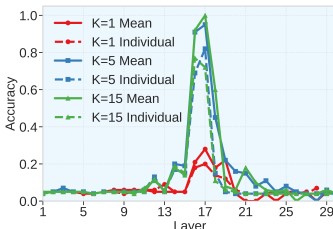 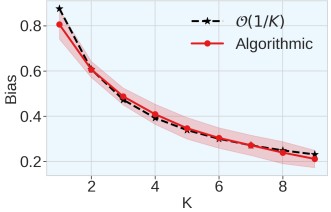 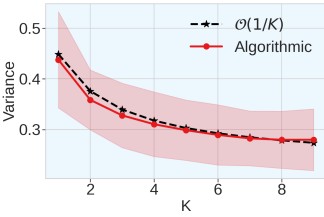

Figure 12: Layerwise task vector accuracy using the mean v.s. individual task vectors.

Figure 13: Decrease of bias at rate of $\mathcal{O}(1/K)$.

Figure 14: Decrease of variance at rate of $\mathcal{O}(1/K)$.

does, whether it does more effectively than individual vectors $\boldsymbol{h}_i(K)$. To address this question, we compare task vector accuracy (Hendel et al., 2023) using both individual task vectors and the mean task vector, with results shown in Figure 12. Remarkably, we observe that injecting the mean task vector consistently leads to better performance, suggesting that it encodes task information more effectively than individual task vectors. Moreover, the performance of the mean task vector also exhibits an inverted U-shaped curve: it peaks around the most compressed layer and improves as the number of demonstrations increases. Together with the results in Figures 10 and 11, this indicates that—ignoring the practical limitations of context length—ICL representations constructed from infinitely many possible demonstrations also exhibit the Layerwise Compression-Expression phenomenon, and the corresponding task vector $\boldsymbol{\mu}(\infty)$ at the most compressed layer can thus be viewed as an ideal task representation. This suggests a promising direction for a theoretical investigation of the phenomenon in the infinite-sample regime, which we leave for future work.

Building on this understanding, we now use the bias–variance decomposition to study how the number of demonstrations $K$ influences task vectors at the most compressed layer.

- **Decrease of bias:** The task mean vector $\boldsymbol{\mu}_t(K)$ progressively shifts from the zero-shot mean vector $\boldsymbol{\mu}_t(0)$ (encodes the pertaining bias) toward the ideal task vector $\boldsymbol{\mu}_t(\infty)$ as $K$ increases, capturing more information about the task. In other words, the bias term $\boldsymbol{\mu}_t(K) - \boldsymbol{\mu}_t(\infty)$ decreases with increasing $K$, and empirically, as shown in Figure 13, we observe that $\|\boldsymbol{\mu}_t(K) - \boldsymbol{\mu}_t(\infty)\|_2 / \|\boldsymbol{\mu}_t(0) - \boldsymbol{\mu}_t(\infty)\|_2$ decays roughly at a rate of $\mathcal{O}(1/K)$.

- **Decrease of variance:** On the other hand, as shown in Figure 14, the variance term $\|\boldsymbol{h}_{i,t}(K) - \boldsymbol{\mu}_t(K)\|_2^2$ decays roughly at a rate of $\mathcal{O}(1/K)$, which together with the fact that between-task distance becomes a constant when $K$ is large leads to a decay rate of $\mathcal{O}(1/K)$ for TDNV.

We remark that, unlike the classical bias–variance decomposition of prediction risk in terms of model capacity—where a trade-off may exist—our bias–variance decomposition applies to the task vector in ICL and exhibits no such trade-off with respect to the number of demonstrations $K$. In other words, both the bias and variance decrease as $K$ increases, indicating that the task vector converges to an ideal task representation. This further suggests that LLMs generate increasingly compact and informative representations from the input demonstrations in the compression phase, with the compact representation encoding task information and converging as the number of demonstrations becomes sufficiently large.

**Theoretical analysis of bias-variance terms for attention layer.** To develop a theoretical understanding of how the bias and variance terms of task vectors evolve with the number of demonstrations $K$, we consider a simplified setting. Specifically, we analyze a single-layer attention model, as the attention mechanism plays a central role in extracting task representations from demonstrations. To simplify the presentation, we drop the subscript $t$ as we only focus on one task and assume that each demonstration and the query correspond to a single hidden state, denoted as $\boldsymbol{h}_1, \ldots, \boldsymbol{h}_K \in \mathbb{R}^d$ for the demonstrations and $\boldsymbol{h}_q \in \mathbb{R}^d$ for the query, respectively. To facilitate analysis, we adopt a linear attention mechanism, denoted by $\mathrm{Attn}$, which preserves the normalization property of softmax attention, namely, that the attention weights sum to 1 (Katharopoulos et al., 2020; Shen et al., 2021). Linear attention has been widely adopted in the literature for theoretical analysis of ICL (Von Oswald et al., 2023; Ahn et al., 2023; Wang et al., 2024; Li et al., 2024).

**Theorem 1.** *Suppose that each demonstration $\boldsymbol{h}_i, i = 1, \ldots, K$ is i.i.d. randomly generated from a distribution $\mathcal{H}$ on $\mathbb{R}^d$. Then the output of the query token, $\boldsymbol{h}_q'(K) = [\mathrm{Attn}(\boldsymbol{h}_1, \ldots, \boldsymbol{h}_K, \boldsymbol{h}_q)]_{:,K+1}$,*

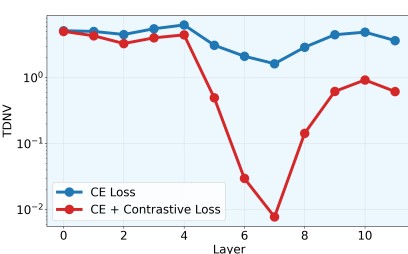

Figure 15: Layerwise TDNV using model trained with CE loss v.s. CE + contrastive loss on layer 7.

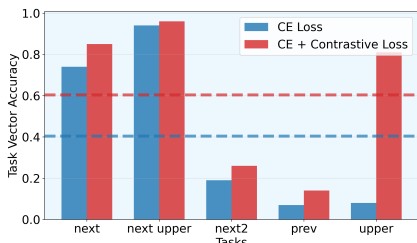

Figure 16: Task-vector contrastive fine-tuning improves task-vector accuracy.

*where $[]_{:,K+1}$ means the $(K+1)$-th column, satisfies the following statistical properties as the number of demonstration $K$ increases:*

- *(**Decrease of variance**) The variance decays as $\mathrm{Var}(\|\boldsymbol{h}'_q(K)\|_2^2) = \mathcal{O}(1/K)$.*
- *(**Decrease of bias**) The mean output $\mathbb{E}[\boldsymbol{h}'_q(K)]$ evolves as a linear combination of the zero-shot mean $\mathbb{E}[\boldsymbol{h}'_q(0)]$ and infinite-shot mean $\mathbb{E}[\boldsymbol{h}'_q(\infty)]$:*

$$\mathbb{E}[\boldsymbol{h}'_q(K)] = \lambda_K \, \mathbb{E}[\boldsymbol{h}'_q(0)] + (1 - \lambda_K) \, \mathbb{E}[\boldsymbol{h}'_q(\infty)], \tag{6}$$

*which further implies that the bias term decays as $\|\mathbb{E}[\boldsymbol{h}'_q(K)] - \mathbb{E}[\boldsymbol{h}'_q(\infty)]\|_2 = \mathcal{O}(1/K)$.*

The proof for Theorem 1 can be found in Appendix E. While our analysis shares the same simplifications and limitations as prior work on linear self-attention (Von Oswald et al., 2023; Ahn et al., 2023; Wang et al., 2024; Li et al., 2024), it offers new insights into the functional role of attention in ICL, revealing how attention contribute to reducing both the variance and bias of task representations—leading to improved performance as the number of demonstrations increases.

## 6 APPLICATIONS OF COMPRESSION-TO-EXPRESSION

**Identify the optimal task vector layer.** To find the optimal intermediate layer for task vector extraction, previous works (Hendel et al., 2023) typically patch the vector at each layer and evaluate accuracy using a validation set. TDNV provides a more efficient way to identify the optimal layer for task vectors. As shown in Figure 1, at a certain intermediate layer $\hat{\ell}$, we observe three simultaneous changes: the TDNV shifts from decreasing to increasing, the task vector accuracy begins to decrease, and the early-exit accuracy starts to increase. Since the layer with minimum TDNV corresponds to the layer with maximum task vector accuracy, we can identify the optimal layer with just one pass of inference using $\hat{\ell} = \arg\min_{\ell \in [1,L]} \mathrm{TDNV}^{(\ell)}$.

**Task-vector contrastive fine-tuning improves task-vector accuracy.** Motivated by prior evidence that more compressed task vectors yield better performance, we propose task-vector contrastive fine-tuning that explicitly encourages such compression. Specifically, during fine-tuning on ICL tasks, we augment the cross-entropy (CE) loss with a contrastive loss applied to the task vectors. This loss pulls representations from the same task closer together while pushing apart those from different tasks (see Appendix H for the exact formulation). We fine-tune a pretrained GPT-2 model on symbolic ICL domains using either the baseline CE loss or our combined loss applied in 7-th layer. As shown in Figure 15, the contrastive term lowers TDNV, indicating stronger task-vector compression, which in turn boosts downstream task-vector accuracy by an average of 20% (Figure 16).

## 7 CONCLUSION

This work provides a comprehensive analysis of the internal dynamics of ICL in LLMs. We uncover a prevalent *Layerwise Compression-Expression* phenomenon in ICL representations, shedding light on how task information is compressed and later expressed to generate predictions. We show that it has profound implications for ICL performance and robustness and reveal the role of attention mechanisms. These insights not only deepen our understanding of structured representation learning in LLMs but also offer practical implications for improving interpretability, efficiency, and robustness.

## ETHICS STATEMENT

This work focuses on revealing the *Layerwise Compression-Expression* phenomenon in in-context learning. Our research does not involve the collection of new human or animal data, and all experiments are conducted using publicly available datasets that have been widely adopted in prior work. We acknowledge that pretrained LLM models may inherit biases present in their training data. Our method involves no additional training of LLMs, it does not explicitly mitigate such biases. We encourage future research to examine fairness, accountability, and transparency when deploying these models in real-world applications.

## REPRODUCIBILITY STATEMENT

We have made every effort to ensure the reproducibility of our work. Full implementation details, including model architectures, hyperparameters, and experimental settings, are provided in the main paper and appendix. To further support reproducibility, we plan to release the complete codebase, configuration files, and detailed instructions upon publication.

## THE USE OF LARGE LANGUAGE MODELS

Large language models were used exclusively to assist with writing polish, grammar correction, and improving readability. They were not used for ideation, experiment design, analysis, or generating research content. All technical contributions, experimental implementations, and results reported in this paper are original work conducted by the authors.

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

The appendix is organized as follows. We first provide detailed descriptions of all tasks in Appendix A. Next, we add more discussion on related works in Appendix B. Appendix C presents additional experiments, including verification of the *Layerwise Compression-Expression*, an ablation study on the choice of the last separation tokens, alternative metrics, multi-hop reasoning tasks and robustness of TDNV. Appendix D offers a fine-grained token-level analysis, featuring saliency maps and grid TDVN visualizations. Proofs of Theorem 1 are given in Appendix E, followed by an examination of the i.i.d. assumption through repetition experiments in Appendix F. We then provide detailed descriptions and illustrations of task-vector accuracy and early-exit accuracy in Appendix G. Finally, Appendix H presents illustrations of the task-vector contrastive fine-tuning method and PCA visualization of extracted task vectors.

## A   TASK DESCRIPTION

### A.1   SYMBOLIC TASKS

Detailed descriptions of the symbolic tasks used in our empirical studies are provided in Table 1, covering the algorithmic, translation, linguistic, and knowledge domains.

Table 1: Descriptions of symbolic tasks.

| Task Domains | Task | Example | Description |
|---|---|---|---|
| Algorithmic(Letter-to-Letter) | Copy Letter | a → a | Output the same letter of the given letter. |
| | Next Letter | a → b | Output the next letter of the given letter in the alphabet. |
| | To Uppercase | a → A | Output the corresponding uppercase letter of the given lowercase letter. |
| | Prev Letter | b → a | Output the previous letter of the given letter in the alphabet. |
| | Next 2 Letter | a → c | Output the letter that comes two positions after the given letter in the alphabet. |
| Algorithmic(List-to-Element) | List First | [a,b,c] → a | Output the first item in the given list. |
| | List Last | [a,b,c] → c | Output the last item in the given list. |
| | List Length | [a,b,c] → 3 | Output length of the given list. |
| | List First Upper | [a,b,c] → A | Get the first item in the given list, then output the corresponding uppercase letter. |
| | List Last Upper | [a,b,c] → C | Get the last item in the given list, then output the corresponding uppercase letter. |
| Translation | French → English | bonjour → hello | Translate the given French word into English. |
| | Spanish → English | gracias → thank you | Translate the given Spanish word into English. |
| | English → French | goodbye → au revoir | Translate the given English word into French. |
| | English → Italian | music → musica | Translate the given English word into Italian. |
| | English → Spanish | thank you → gracias | Translate the given English word into Spanish. |
| Linguistic | Antonyms | hot → cold | Output the antonym of the given word. |
| | Plural → Singular | cats → cat | Convert the given plural noun to its singular form. |
| | Present Simple → Gerund | run → running | Convert the given verb from present simple to its gerund form. |
| | Present Simple → Past Perfect | walk → had walked | Convert the given verb from present simple to past perfect tense. |
| | Present Simple → Past Simple | jump → jumped | Convert the given verb from present simple to past simple tense. |
| | Singular → Plural | dog → dogs | Convert the given singular noun to its plural form. |
| Knowledge | Country → Capital | France → Paris | Output the capital city of the given country. |
| | Football Player → Position | Lionel Messi → Forward | Output the playing position of the given football player. |
| | Location → Continent | Brazil → South America | Output the continent where the given location is found. |
| | Location → Country | Kyoto → Japan | Output the country in which the given location is situated. |
| | Location → Language | Egypt → Arabic | Output the primary language spoken in the given location. |
| | Location → Religion | India → Hinduism | Output the predominant religion of the given location. |

### A.2   LANGUAGE UNDERSTANDING TASKS

For evaluating TDNV on natural language understanding tasks, we require each query sentence to be assessed across multiple attributes. To enable this, we construct a synthetic natural language dataset in which every sentence can be evaluated on six distinct attributes: length, semantic polarity, tense, sentence type, subject person, and entity type. Each attribute is associated with several

categorical labels, all of which are summarized in Table 2. The dataset contains 1,000 samples, and representative examples of these sentences are provided in Table 3.

| Attribute | Labels |
|---|---|
| Length | short, medium, long |
| Semantic Polarity | positive, negative, neutral |
| Tense | present, past, future, progressive |
| Sentence Type | declarative, interrogative, imperative, exclamatory |
| Subject Person | first_person, second_person, third_person |
| Entity Type | person, location, organization |

Table 2: The attributes and labels in language understanding dataset.

| Sentence | Length | Semantic Polarity | Tense | Sentence Type | Subject Person | Entity Type |
|---|---|---|---|---|---|---|
| I enjoy morning walks. | short | positive | present | declarative | first_person | person |
| Close the window now. | short | neutral | present | imperative | second_person | location |
| Despite the heavy rain, our research team successfully completed the outdoor experiment and gathered all the required samples before sunset. | long | positive | past | declarative | third_person | organization |
| Will you be visiting the United Nations headquarters in New York next year to attend the global climate summit? | long | neutral | future | interrogative | second_person | location |
| While the orchestra rehearsed the challenging new symphony, the conductor meticulously adjusted each section to achieve the perfect balance of sound for the upcoming performance. | long | neutral | progressive | declarative | third_person | organization |

Table 3: Example sentences in language understanding dataset, each sentence is annotated with 6 attributes.

## A.3 MULTIMODALITY TASKS

For evaluating TDNV on multimodality tasks, we require each query image to be assessed across multiple attributes. To enable this, we construct a synthetic vision-text dataset in which every image can be evaluated on four distinct attributes: color, shape, size and texture. Each attribute is associated with several categorical labels, all of which are summarized in Table 4. The dataset contains 300 samples, and representative examples of these images are provided in Figure 17.

| Attribute | Labels |
|---|---|
| Color | red, green, blue, yellow, black |
| Shape | circle, square, triangle, pentagon, star |
| Size | small, medium, large |
| Texture | solid, stripes, dots, checker |

Table 4: The attributes and labels in multimodality dataset.

## B RELATED WORKS

**Layerwise Representations** An intriguing line of research (Ben-Shaul & Dekel, 2022; Fang et al., 2021; Wang et al., 2023b; Rangamani et al., 2023; He & Su, 2024; Zhou et al., 2025) has empirically investigated the role of different layers in feature learning. These studies show that in image classification tasks, features in intermediate layers become increasingly linearly separable as the layers deepen. Specifically, Neural Collapse ($\mathcal{NC}$) properties emerge in intermediate layers, where the within-class variance decreases compared to the between-class variance as depth increases. This indicates that layerwise compression occurs **monotonically** with layer depth in these settings. However, our hypothesis reveals that in the ICL setting, decoder-only models' layerwise representations exhibit **dual** encoding-decoding stages: shallow layers compress information while deep layers express it. Furthermore, research by (Skean et al., 2025) shows that intermediate layers consistently outperform both shallow and final layers on downstream tasks. It analyzes information compression

| image | color | shape | size | texture |
|---|---|---|---|---|
|  | yellow | triangle | small | solid |
|  | blue | square | large | dots |
|  | red | star | large | solid |
|  | black | pentagon | large | checker |
|  | green | circle | small | stripes |

Figure 17: Example images in multimodality dataset, each image is annotated with 4 attributes.

through the intra-sequence geometry of tokens within a single prompt (e.g., measuring curvature and entropy). In contrast, our work investigates inter-instance compression at the task level. Specifically, we measure how representations of distinct ICL instances belonging to the same task cluster together while separating from different tasks, rather than analyzing the local distribution of token embeddings within a sequence.

**In-Context Learning Interpretability** Numerous studies have focused on the working mechanisms of ICL (Xie et al., 2021; Chen et al., 2021; Von Oswald et al., 2023; Dai et al., 2022; Ahn et al., 2023; Olsson et al., 2022). Xie et al. (2021) propose that ICL emerges as an implicit form of Bayesian inference. In the realm of meta-learning, Chen et al. (2021) introduce in-context tuning to predict target labels from concatenated sequences of task instructions. A significant line of research connects ICL with gradient descent optimization. Von Oswald et al. (2023) demonstrate that Transformers trained on autoregressive tasks can emulate gradient descent in their forward pass. Dai et al. (2022); Ahn et al. (2023) compare standard gradient descent-based fine-tuning with ICL, revealing that transformer attention in ICL exhibits a dual form of gradient descent-based optimization. Olsson et al. (2022) identify "induction heads"—specific attention heads that facilitate ICL by copying patterns from previous tokens. However, our work focuses on the layer-wise representational analysis of ICL. In addition, Doimo et al. (2024) also reveals different clustering patterns in shallow and deep layers. However, we observed a strikingly different phenomenon and would like to clarify the differences in settings and results. Doimo et al. (2024) uses demonstrations with minimal task cues while the query itself provides ample task information, enabling high zero-shot accuracy and only modest few-shot gains. Our tasks encode the task only in the demonstration pairs, while the query provides no clues, yielding near-random zero-shot performance. As a result, our work shows that the model progressively compresses task information from the demonstration pairs, reaching maximal compression around the intermediate layers. In contrast, Doimo et al. (2024) clusters representations by semantic subject, with ARI peaking in the earliest layers.

**Task Representations** Researchers have explored various ways to extract compact representations of different ICL tasks, including task vectors (Hendel et al., 2023), function vectors (Todd et al., 2023), and state vectors (Li et al., 2024). Task vectors (Hendel et al., 2023) are extracted from the intermediate hidden state of the final separate token. Function vectors (Todd et al., 2023) are derived from attention activation through causal mediation, while state vectors (Li et al., 2024) concatenate activations from several initial layers of transformers. These representations effectively enable models to perform ICL tasks by injecting to specific transformer layers' hidden states during inference. Some researchers have explored task manipulation within in-context learning. For instance, Shao et al. (2023) has demonstrated that compositional task representations can be created through composition model training. Additionally, In-context vectors (Liu et al., 2023b) enhance ICL through latent space steering. However, previous works have mainly focused on task representations for individual tasks, and none have provided a layer-wise analysis of task vectors. Our research examines how the model distinguishes between different tasks from a geometric perspective across shallow to deep layers.

# C ADDITIONAL EXPERIMENTS

## C.1 UNIVERSAL ACROSS DIFFERENT TASKS AND MODELS

In Figure 1, we validate the *Layerwise Compression-Expression* phenomenon across layers on Letter-to-Letter tasks. To further assess its generality, we evaluate the phenomenon on the other algorithmic task groups: (i) List-to-Element tasks (list-first, list-last, list-length, list-first-upper, list-last-upper); (ii) A combination of Letter-to-Letter and List-to-Element tasks.

We use the DeepSeek-Coder-7B model under a 15-shot ICL setting. As shown in Figure 18, the TDNV exhibits a U-shaped curve across layers in both settings, and both task vector accuracy and early-exit accuracy follow patterns similar to those observed in Figure 1. Notably, Figure 18(b) presents results for the combined task groups listed in Table 1, further supporting the conclusion that this phenomenon holds broadly across diverse tasks. These findings further confirm the universality of the *Layerwise Compression-Expression*.

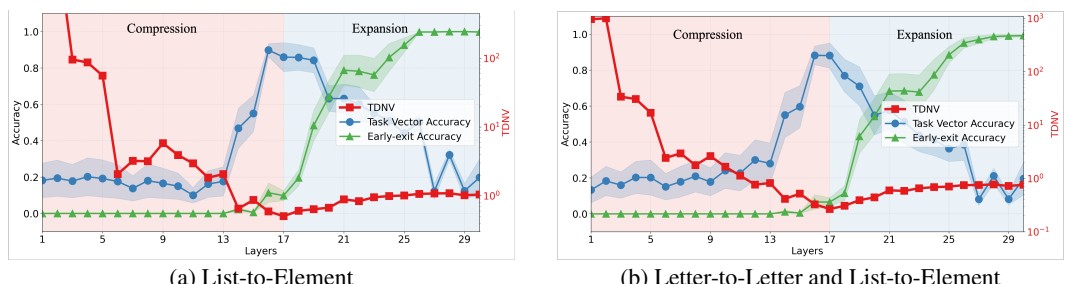

(a) List-to-Element          (b) Letter-to-Letter and List-to-Element

Figure 18: *Layerwise Compression-Expression* phenomenon across different tasks groups: a) List-to-Element, b) a combination of two task groups, the Letter-to-Letter and List-to-Element. TDNV first decreases then increases from shallow to deep layers, splitting the model into compression and expression stages.

To further validate our findings, we extended our evaluation to include Gemma2-9B (Team et al., 2024) and Mistral-7B (Jiang et al., 2023a) on symbolic ICL tasks using 15-shot, measuring layerwise TDNV, task vector accuracy, and early-exit accuracy. As illustrated in Figure 19, we observe a clear U-shaped TDNV pattern that aligns perfectly with our previous results. When combined with the five models originally analyzed (Deepseek-coder, GPT-J, Llama, Pythia, and Mamba) in Figure 2, we have now demonstrated consistent behavior across seven distinct architectures.

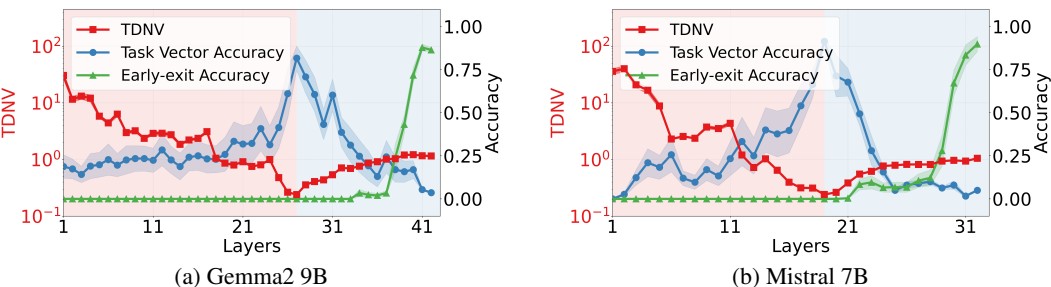

(a) Gemma2 9B          (b) Mistral 7B

Figure 19: *Layerwise Compression-Expression* phenomenon on Gemma2 9B and Mistral 7B models on symbolic ICL tasks group. TDNV first decreases then increases from shallow to deep layers, splitting the model into compression and expression stages.

## C.2 ALTERNATIVE REPRESENTATIONS FOR TASK VECTOR

We choose the last separator token as the ICL representation following prior work (Hendel et al., 2023), where it serves as a natural anchor point between the demonstrations and the query. For comparison, we evaluate two other aggregation strategies: ($i$) Mean of All Tokens: the mean of all demonstration token representations(remove the last query and separator), and ($ii$) Mean of All Separator Tokens: the mean of all separator token representations. To quantitatively evaluate which

representation best captures layerwise features, we compare the TDNV changes during both compression and expression stages:

$$\Delta_{\text{Compression}} = \frac{\text{TDNV}_0 - \text{TDNV}_{\ell_{\min}}}{\text{TDNV}_0} \qquad \text{and} \qquad \Delta_{\text{Expression}} = \frac{\text{TDNV}_L - \text{TDNV}_{\ell_{\min}}}{\text{TDNV}_L}.$$

Where $\text{TDNV}_0$ and $\text{TDNV}_L$ are the TDNV of the first and last layer, and $\text{TDNV}_{\ell_{\min}}$ is the minimum TDNV.

| Representation | Compression Ratio ↑ | Expression Ratio ↑ |
|---|---|---|
| Mean All Tokens | 0.9671 | 0.6002 |
| Mean Sep Tokens | 0.9879 | 0.5448 |
| **Last Sep Token** | **0.9926** | **0.7746** |

Table 5: Compression and expression ratios for different representations.

As shown Table 5, the Last Sep Token representation demonstrates both higher compression and expression ratios. This evidence supports our conclusion that the last separator token remains the optimal choice for capturing task-relevant information.

The other two alternatives are suboptimal for the following reasons:

- Mean of All Tokens. As shown in Figure 24, saliency maps reveal that token contributions to task representation are highly uneven. Early layers focus on label tokens within the demonstrations, while later layers shift attention to the final separator token as the primary carrier of task information. Averaging across all tokens therefore introduces noise from irrelevant content tokens and dilutes the in-context learning (ICL) signal.

- Mean of All Separator Tokens. As illustrated in Figure 25, grid-level TDNV analysis across separator tokens shows a monotonic decrease in TDNV from the first to the last separator. This pattern indicates that the model progressively compresses task-relevant information across successive demonstrations. Because later separators encode richer context and stronger compression, averaging over all separators weakens this effect, whereas using only the final separator captures the fully accumulated task representation.

## C.3 Alternative Metrics

To validate the robustness of our geometric analysis, we compare our proposed Task-Distance Normalized Variance (TDNV) against alternative clustering metrics, including nearest-class-center (NCC) variance and the silhouette score. While NCC variance measures class compactness, we find that it fails to fully capture the distinct "compression-expression" mechanism observed with TDNV, as it neglects the crucial aspect of inter-task separation (see Figure 20). Similarly, although the silhouette score assesses general clustering quality based on sample-to-sample distances, it does not explicitly model the geometry of class centroids (means). This distinction is critical for ICL, where the "task vector" is widely conceptualized as the mean representation of demonstrations (Hendel et al., 2023). By focusing on the separability of these task centroids relative to their variance, TDNV offers a more direct physical interpretation of how robustly the task definition has been extracted.

To quantitatively validate the superiority of TDNV over these alternatives, we evaluated how well each metric correlates with actual downstream model performance. Specifically, we measured the statistical dependence between layerwise geometric scores (TDNV, NCC variance, and silhouette score) and task vector accuracy using distance correlation (dCor). Unlike standard Pearson correlation, dCor is a robust statistical measure capable of detecting both linear and non-linear dependencies, where 0 indicates independence and 1 indicates strong dependency. As shown in Table 6, TDNV exhibits a stronger correlation with task vector accuracy compared to alternative metrics across both Mistral-7B and Gemma2-9B models. These consistently higher dCor values confirm that TDNV is the better predictor of model performance. Unlike generic clustering metrics, TDNV accurately reflects the 'quality' of the task information extracted by the model, directly linking the geometric structure of the representation to the model's actual ability to solve the task.

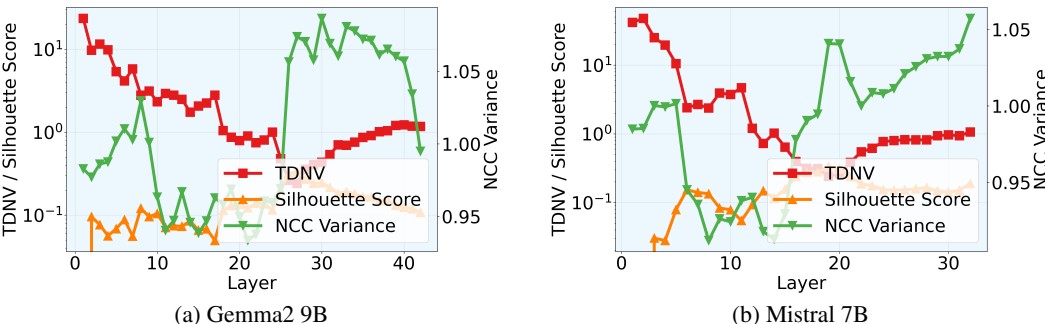

(a) Gemma2 9B          (b) Mistral 7B

Figure 20: Comparison of TDNV with other geometric metrics on Gemma2 9B and Mistral 7B. Layerwise representation is extracted on symbolic ICL task group.

Table 6: Distance Correlation (dCor) between layerwise geometric metrics and task vector accuracy. TDNV consistently shows the strongest dependency with downstream performance.

| Model | NCC Variance | Silhouette Score | TDNV (Ours) |
|---|---|---|---|
| Mistral-7B | 0.3436 | 0.7297 | **0.7539** |
| Gemma2-9B | 0.4265 | 0.7962 | **0.8558** |

## C.4 CHAIN-OF-THOUGHT MULTI-HOP REASONING

To examine the Layerwise Compression-Expression within reasoning contexts, we designed a suite of 2-hop chain-of-thought tasks by composing elementary operations such as *Next Letter*, *Previous Letter*, *To Uppercase*, and *Copy Letter*. Each instance follows a chain-of-thought structure formatted as "input $\rightarrow$ intermediate_step $\rightarrow$ final_output" (e.g., $a \rightarrow b \rightarrow B$, representing *Next Letter* followed by *To Uppercase*). Crucially, to capture the temporal evolution of task processing, we compute the layerwise TDNV at two distinct anchor points: the separator token preceding the intermediate output (Hop 1) and the separator preceding the final output (Hop 2). As shown in Figure 22, our analysis reveals that the *Layerwise Compression-Expression* phenomenon exhibits recursive pattern: we observe multiple distinct cycles corresponding to each reasoning hop. Specifically, the model exhibits a full compression-expression trajectory to generate the intermediate output (Hop 1) and subsequently initiates a new cycle to process that intermediate result for the final output (Hop 2). This suggests that LLMs decompose multi-hop ICL tasks into cascaded compression-expression operations rather than compressing the entire reasoning chain into a single global representation.

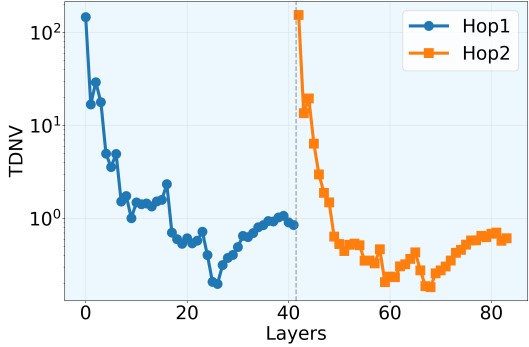

Figure 21: TDNV exhibits compression-expression cycles in chain-of-thought multi-hop reasoning.

## C.5 ROBUSTNESS OF TDNV

We conducted extensive ablation studies to verify the stability of our proposed metric. As shown in Figure 22, the TDNV metric proves highly robust to experimental design choices, maintaining

a consistent U-shaped curve with negligible variance across varying conditions. Specifically, we observe the same layerwise trend regardless of the distance metric employed (L1 vs. L2 norm), the number of samples used for estimation ($N = 100$ or $200$), or the specific prompt format (using "→" vs. ":"). This stability confirms that the observed *Layerwise Compression-Expression* phenomenon is an intrinsic property of the model's representational dynamics rather than an artifact of specific hyperparameters or measurement configurations.

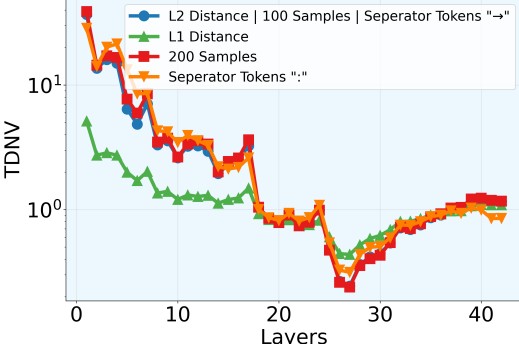

Figure 22: Robustness of TDNV metric under different distance metrics, sample sizes, and prompt formats.

## D  TOKEN LEVEL ANALYSIS

### D.1  SALIENCY MAPS

In the main pages, we have extensively explored the use of TDNV to quantify layerwise information compression during ICL, revealing important statistical properties of model representations. In this subsection, we complement the understanding from a fine-grained token level. In particular, we will use the method of saliency maps (Wang et al., 2023a), specifically elucidating which parts of the input significantly contribute to the model's decision-making. By highlighting critical token interactions, saliency maps provide intuitive insights into model behavior. Denoting by $I_\ell$ saliency map at the $\ell$-th layer, we compute it by,

$$I_\ell = \left| \sum_h A_{h,\ell} \odot \frac{\partial \mathcal{L}}{\partial A_{h,\ell}} \right|, \tag{7}$$

where $A_{h,\ell}$ represents the attention map of head $h$ at layer $\ell$, and the loss $\mathcal{L}$ is the cross-entropy calculated between the logits of the last token and the ground-truth label. Thus, a saliency map quantifies the importance of internal activations by integrating both attention strength and its gradient-based influence on the model's outcome. In a nutshell, these maps highlight how token interactions evolve across layers.

We show saliency maps of all layers using three demonstrations in Figure 23. In shallow layers, there is more interaction within demonstrations, indicating that the model extracts task information from each demonstration. In deep layers, there is less interaction within demonstrations and more interaction with the last token, indicating that the model uses the accumulated task information to generate output.

To observe more clear patterns, we group the layers into 3 groups: shallow (average of layers 1-14), intermediate (average of layers 15-16), and deep (layers 17 and above) in Figure 24. We observe that $(i)$ in shallow layers, the model focuses on token-to-label mappings within demonstrations, reflecting local compression, $(ii)$ intermediate layers shift focus toward the final token, indicating integration of task-level information, and $(iii)$ deep layers emphasize interactions between the query and final token, aligning with expression and prediction. Thus, token-level interpretability also aligns with the layerwise compression-expression trajectory.

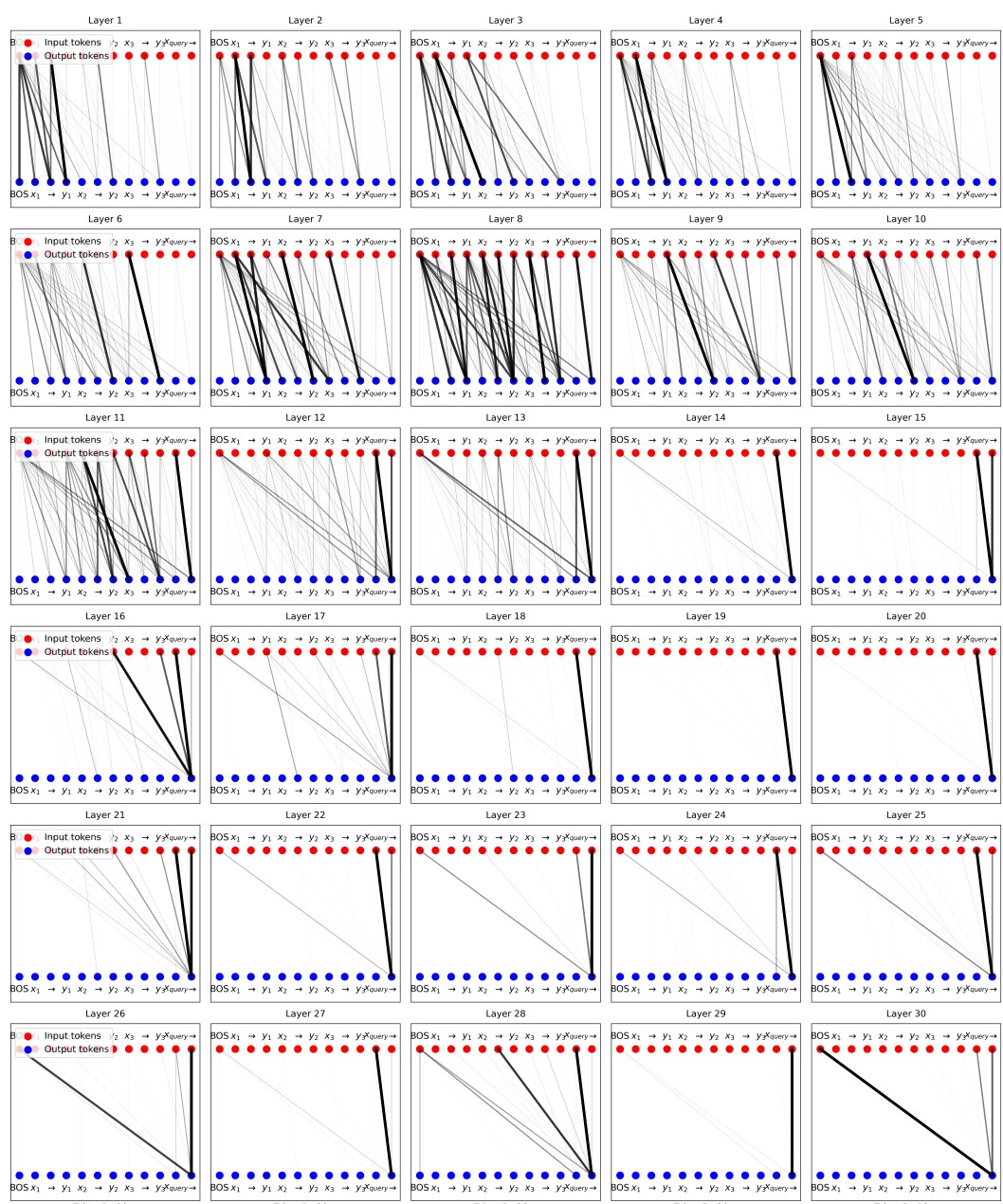

Figure 23: Saliency Maps for all layers.

## D.2 GRID-TDNV

In Section 4.3, we find that the position of noisy demonstration affects compression. To investigate why perturbing later demonstrations results in higher TDNV at the final separation token, we conduct a more fine-grained token-level analysis by computing the layerwise TDNV for each separation token across all demonstrations. For each demonstration's separator, we calculate the layerwise TDNV and organize them into a grid structure, referred to as the grid TDNV. As shown in Figure 25, perturbing a demonstration at a given position most significantly increases the TDNV of the next demonstration. However, this increase gradually diminishes as more correct demonstrations are appended. This pattern suggests that the negative impact of early errors can be partially mitigated by subsequent correct examples.

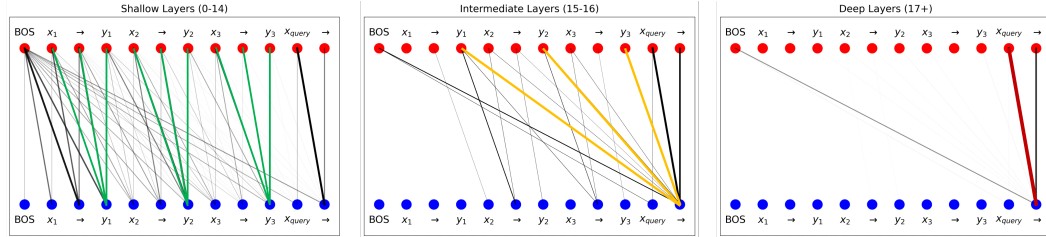

Figure 24: Saliency maps across transformer layers: (a) shallow, (b) intermediate, and (c) deep. The edge widths indicate saliency magnitude from input tokens (red dots) to output tokens (blue dots).

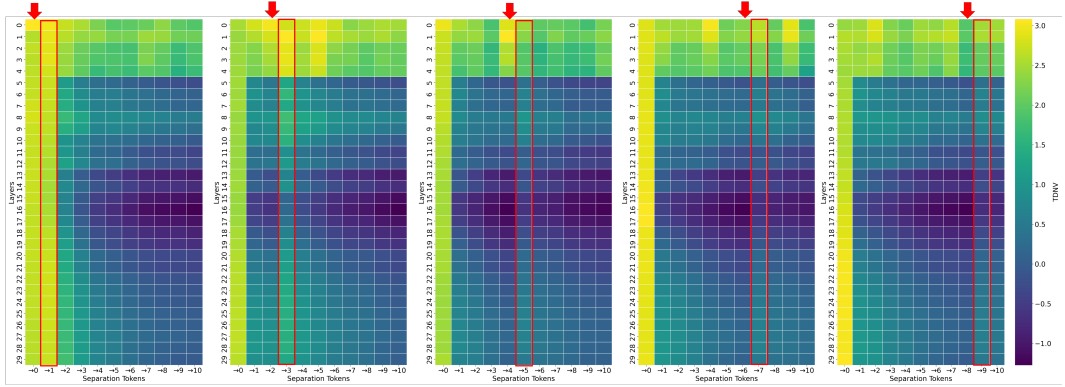

Figure 25: The grid TDNV pattern when perturbing one demonstration at different positions: from left to right, the perturbation is applied to the 0th, 2nd, 4th, 6th, and 8th demonstration.

## E  PROOF OF THEOREM 1

*Proof.* (**Proof for Variance Decay.**) Consider one layer linear attention (Von Oswald et al., 2023; Ahn et al., 2023; Wang et al., 2024; Li et al., 2024) that preserves the normalization property of softmax attention(Katharopoulos et al., 2020; Shen et al., 2021): the hidden state of the query token becomes

$$\boldsymbol{h}'_q(K) = [\text{Attn}(\boldsymbol{h}_1, \ldots, \boldsymbol{h}_K, \boldsymbol{h}_q)]_{:,K+1}$$

$$= \sum_{i=1}^{K} \frac{\phi(\boldsymbol{q}_q)^\top \phi(\boldsymbol{k}_i)}{K+1} \boldsymbol{v}_i + \frac{\phi(\boldsymbol{q}_q)^\top \phi(\boldsymbol{k}_q)}{K+1} \boldsymbol{v}_q, \tag{8}$$

where $\boldsymbol{q} = \boldsymbol{W}^Q \boldsymbol{h}, \boldsymbol{k} = \boldsymbol{W}^K \boldsymbol{h}, \boldsymbol{v} = \boldsymbol{W}^V \boldsymbol{h}$ are the query, key and value vectors, respectively, and $\phi : \mathbb{R}^d \to \mathbb{R}^r$ denotes the feature map that approximates the softmax. Define $z_j := \phi(\boldsymbol{q}_q)^\top \phi(\boldsymbol{k}_j)$, $z_q := \phi(\boldsymbol{q}_q)^\top \phi(\boldsymbol{k}_q)$, we can rewrite it as

$$\boldsymbol{h}'_q(K) = \sum_{i=1}^{K} \frac{z_i}{K+1} \boldsymbol{v}_i + \frac{z_q}{K+1} \boldsymbol{v}_q. \tag{9}$$

To further simplify the notation, we define $\boldsymbol{a}_i := z_i \boldsymbol{v}_i, \quad i = 1, \ldots, K$, and $\boldsymbol{a}_{K+1} := z_q \boldsymbol{v}_q$, which gives

$$\boldsymbol{h}'_q(K) = \frac{1}{K+1} \sum_{i=1}^{K+1} \boldsymbol{a}_i. \tag{10}$$

Since $\boldsymbol{a}_i := \phi(\boldsymbol{q}_q)^\top \phi(\boldsymbol{W}^K \boldsymbol{h}_i) \boldsymbol{W}^V \boldsymbol{h}_i$ only relates to $\boldsymbol{h}_i$ and $\boldsymbol{h}_i$ are i.i.d., the $\boldsymbol{a}_i$ are also i.i.d. with

$$\mathbb{E}[\boldsymbol{a}_i] = \boldsymbol{\mu}_a, \qquad \text{Cov}(\boldsymbol{a}_i) = \boldsymbol{\Sigma}_a.$$

Finally, we get,

$$\mathrm{Var}(\|\boldsymbol{h}_q'\|_2^2) = \frac{1}{(K+1)^2} \sum_{i=1}^{K+1} \mathrm{Var}(\|\boldsymbol{a}_i\|_2^2) = \frac{K+1}{(K+1)^2} \mathrm{Tr}(\boldsymbol{\Sigma}_a) = \frac{\mathrm{Tr}(\boldsymbol{\Sigma}_a)}{K+1} \sim \mathcal{O}(1/K). \quad (11)$$

$\square$

*Proof.* (**Proof for Mean Shift.**) Define the mean of demonstrations and the zero-shot as

$$\boldsymbol{\mu}_{\text{zero-shot}} := \mathbb{E}[z_q \boldsymbol{v}_q], \qquad \boldsymbol{\mu}_{\text{demo}} := \mathbb{E}[z_i \boldsymbol{v}_i].$$

Then according to (9), we have

$$\mathbb{E}[h_q'(K)] = \frac{1}{K+1} \underbrace{\mathbb{E}[z_q \boldsymbol{v}_q]}_{=:\boldsymbol{\mu}_{\text{zero-shot}}} + \frac{K}{K+1} \underbrace{\mathbb{E}[z_i \boldsymbol{v}_i]}_{=:\boldsymbol{\mu}_{\text{demo}}}. \quad (12)$$

If $K = 0$, we get $\mathbb{E}[\boldsymbol{h}_q'(0)] = \mathbb{E}[z_q \boldsymbol{v}_q]$. When $K \to \infty$, we have $\mathbb{E}[\boldsymbol{h}_q'(\infty)] = \mathbb{E}[z_i \boldsymbol{v}_i]$.

In summary, we get,

$$\mathbb{E}[\boldsymbol{h}_q'(K)] = \lambda_K \, \mathbb{E}[\boldsymbol{h}_q'(0)] + (1 - \lambda_K) \, \mathbb{E}[\boldsymbol{h}_q'(\infty)] \quad (13)$$

where $\lambda_K = 1/(1+K) \sim \mathcal{O}(1/K)$.

$\square$

# F    VALIDATION OF THE I.I.D. ASSUMPTION UNDER REPITITION

In Theorem 1, we assume that each demonstration $\boldsymbol{h}_i, i = 1, \ldots, K$ is i.i.d. randomly generated from a distribution $\mathcal{H}$ on $\mathbb{R}^d$. To validate the importance of this assumption, we design experiments that increase the ICL length by providing additional demonstrations as input. There are two possible extension methods:

- Repeat Mode: Extending demonstrations by repeating existing examples.
- Distinct Mode: Extending demonstrations by adding new, unique examples.

In the Distinct Mode, new demonstrations are independently and identically distributed (i.i.d.), randomly generated from a distribution. In the Repeat Mode, demonstrations are no longer i.i.d. We examine both performance and compression level across these two modes.

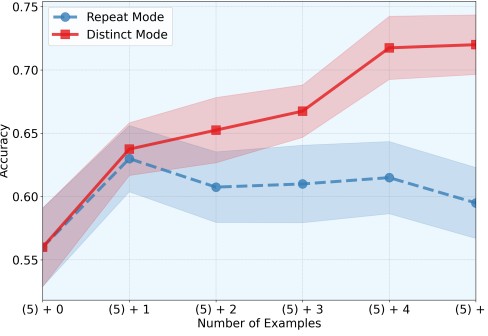

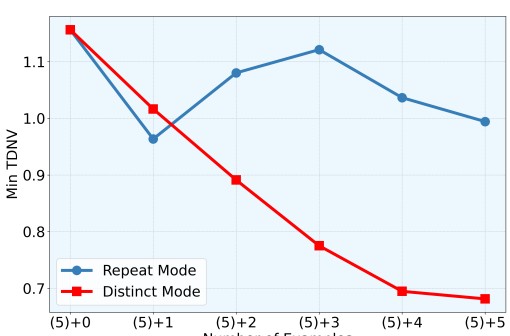

Figure 26: Comparison of accuracy under repeat mode v.s. distinct mode.

Figure 27: Minimum TDNV under under repeat mode v.s. distinct mode.

As shown in Figure 26 and Figure 27, in distinct mode, as new demonstrations introduce novel information, the accuracy increases and TDNV decreases. However, in repeat mode, since repeated demonstrations add no new task information, the accuracy and TDNV remain largely unchanged. This proves that when the i.i.d. condition is violated, increasing the number of demonstrations does not lead to better performance and compression.

## G    TASK VECTOR ACCURACY & EARLY-EXIT ACCURACY

Task vector accuracy(Hendel et al., 2023) refers to how accurately a LLM can perform a task using only a learned representation of the task(task vector), instead of full in-context demonstrations. As shown in Figure 28, to evaluate the task vector accuracy, we conduct the following steps:

1. **Extract** task vector $\theta$ from the demonstration set $\mathcal{S}_K$ using a dummy query $x'$, avoiding information leakage from the real query as,

$$\theta_{\text{task}} = [f_{\theta^{(1:\ell)}}([\mathcal{S}_K, x'])]_{:,K+1} \tag{14}$$

2. **Inject** $\theta$ into a forward pass of the model with only the query $x$, not the full demonstration set, then predict the output using this modulated forward pass.

$$y = f_\theta([x]; \theta_{task}) \tag{15}$$

where $[]_{:,K+1}$ means the $(K+1)$-th column, $\ell$ is the layer with most compression.

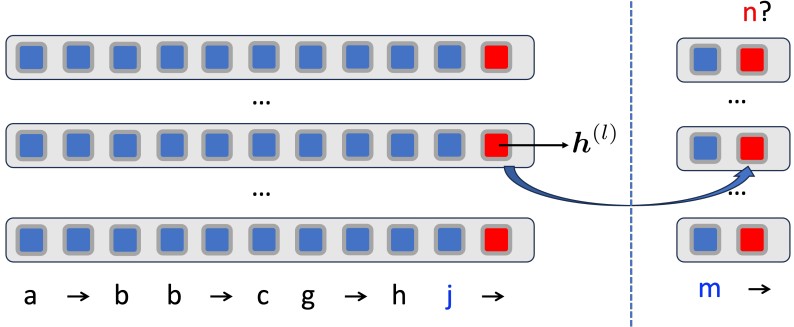

Figure 28: Illustration for task vector accuracy.

Early-exit accuracy (Xin et al., 2020; Jiang et al., 2024a) measures a model's prediction accuracy when using intermediate-layer representations instead of the final layer for making predictions. This metric helps assess how effectively different layers encode the information needed to complete the task. Let $h^{(\ell)} \in \mathbb{R}^d$ be the hidden state of the final token at layer $\ell$, and let $\mathcal{C} : \mathbb{R}^d \to \mathbb{R}^V$ be the last-layer classifier mapping from hidden dimension $d$ to vocabulary size $V$. Then as shown in Figure 29, the prediction is,

$$\hat{y}^{(\ell)} = \arg\max_{v \in \mathcal{V}} \text{softmax}(\mathcal{C}(\mathbf{h}^{(\ell)}))_v \tag{16}$$

Then, the early-exit accuracy at layer $\ell$ over $N$ examples is,

$$\text{Acc}^{(\ell)} = \frac{1}{N} \sum_{i=1}^{N} \mathbf{1}\left[\hat{y}_i^{(\ell)} = y_i\right] \tag{17}$$

## H    TASK-VECTOR CONTRASTIVE FINE-TUNING

As shown in Figure 30, to obtain more compressed and discriminative task vectors, we add an additional contrastive loss on task vectors that explicitly encourages representation compression. Specifically, we fine-tuning a pretrained GPT-2 model on algorithmic ICL tasks domains (next letter, next two letters, previous letter, uppercase, next uppercase letter) using a combined objective as,

$$\mathcal{L} = \mathcal{L}_{\text{CE}} + \beta \mathcal{L}_{\text{Contra}}, \tag{18}$$

where $\mathcal{L}_{\text{CE}}$ is the cross-entropy loss computed only on separator tokens to predict the correct label and $\mathcal{L}_{\text{Contra}}$ is a contrastive term that pulls hidden states from the same task closer while pushing

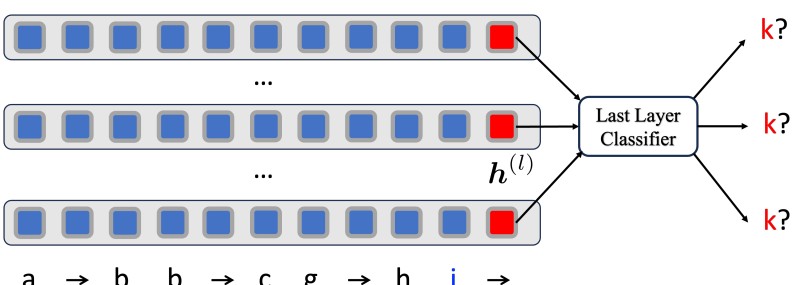

Figure 29: Illustration for early-exit accuracy.

those from different tasks apart:

$$\mathcal{L}_{\text{Contra}} = -\frac{1}{|\Omega|} \sum_{t} \sum_{\substack{i,j \\ i \neq j}} \log \frac{\exp(\mathbf{h}_{i,t}^{\top} \mathbf{h}_{j,t}/\tau)}{\sum_{(a,b) \neq (i,t)} \exp(\mathbf{h}_{i,t}^{\top} \mathbf{h}_{a,b}/\tau)}. \tag{19}$$

where $\mathbf{h}_{i,t}$ denotes the normalized hidden state of the last token in the intermediate layer(task vector) for the $i$-th sample of task $t$, $\tau$ is the temperature, and $\Omega$ is the set of all pairs. We set $\beta = 0.1$, temperature $\tau = 0.07$, and batch size 100 with equal samples from each task. Each training example provides $K = 20$ context demonstrations followed by a separator token. We finetuned the model to achieve 100% ICL accuracy on all tasks.

For evaluation, we extract task vectors with number of demonstrations $K = 20$ and assess task-vector accuracy: performing zero-shot ICL with the extracted task vector injected at the same position. Higher task-vector accuracy reflects more effective task vectors. As shown in Figure 16, task-vector contrastive fine-tuning produces better task vectors than standard fine-tuning.

To illustrate whether task-vector contrastive fine-tuning produces more compressed task vectors, we visualize hidden states from the 7-th layer using PCA. As shown in Figure 31, the left panel depicts task vectors extracted from model finetuned with only the cross-entropy loss, while the right panel shows vectors extracted from model finetuned with both cross-entropy and contrastive losses. The latter exhibits more distinct and tightly clustered task representations, demonstrating the effectiveness of the contrastive objective in compressing task vectors.

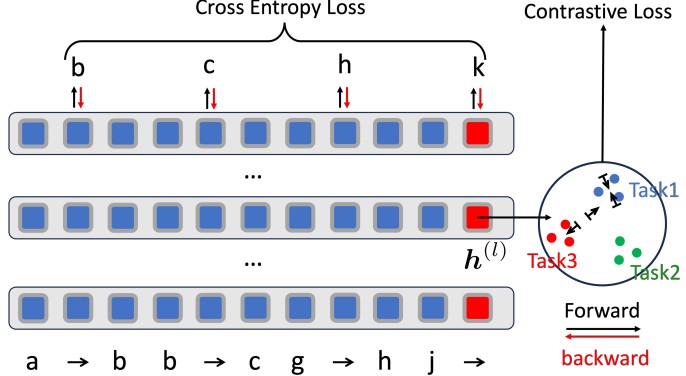

Figure 30: Illustration of task-vector contrastive fine-tuning.

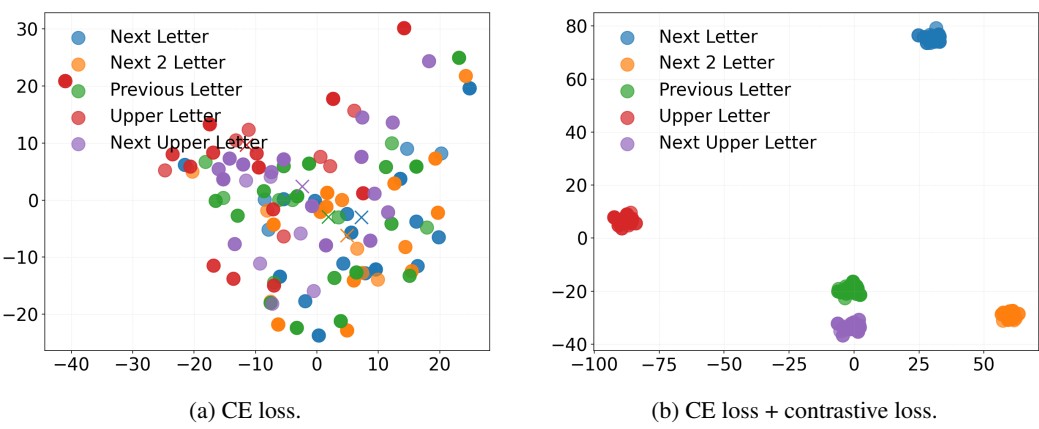

(a) CE loss.

(b) CE loss + contrastive loss.

Figure 31: Comparison of PCA visualizations for task vectors. The task vector is extracted from models finetuned with different losses.

