# OpenReview forum: "From Compression to Expression: A Layerwise Analysis of In-Context Learning"
_ICLR.cc/2026/Conference — Submitted to ICLR 2026_

### Official Review · Reviewer_nSwE · 2025-10-24

**Soundness:** 1
**Presentation:** 3
**Contribution:** 1
**Rating:** 2
**Confidence:** 5

**Summary:**

This paper proposes the "layerwise" spread of information compression in ICL, where early layers compress task information and later layers express it. The authors' proposed approach TDNV (Task-Distance Normalized Variance) quantifies compression and observe U-shaped patterns across models and tasks. While the empirical scope is extensive, concerns about novelty relative to recent work on layerwise information dynamics, generalizability across modern architectures, and lack of statistical rigor in the experiments limit the contribution.

**Strengths:**

- Comprehensive empirical evaluation across architectures and modalities
- Clear presentation with intuitive visualizations
- Practical applications such as TDNV-based layer selection

**Weaknesses:**

### **Major**

**Generalizability and credibility of the findings:** The motivation and method appear very similar to Skean et al.'s work [1], which studies information compression across layers with variance-based technical analysis. The authors essentially replicate this using the same models. Studying ICL in this context offers limited novelty, as ICL is merely one instance of task/information encoded in text.

Furthermore, recent work [2] demonstrates that information compression doesn't necessarily follow a U-shaped pattern, despite common claims [1, 3, 4]. Saglam et al. [2] show that while the U-shaped pattern exists in models tested here (Llama, GPT), it doesn't hold in others like Mistral and Gemma. This indicates information compression patterns heavily depend on model architecture, rendering the paper's findings questionable.

The authors observe U-shaped patterns across all tested models. Without counter-evidence, they should expand their model selection to ensure academic rigor. So I don't think this work is sufficiently concrete and novel for publication.

**Causal vs. geometric interpretation:** The paper claims compression "extracts task information from demonstrations" and expression "applies to the query," but TDNV measures geometric properties (clustering, separation), not causal information flow. Saliency maps (Appendix D.1) provide suggestive evidence, but more rigorous connections are needed:
- information-theoretic analysis (e.g., mutual information between demonstrations and task vectors),
- causal interventions (e.g., ablating early vs. late layers and measuring downstream effects),
- or explicit framing of geometric patterns as correlates rather than causes of information compression.

**Statistical rigor in experiments:** Results lack error bars, confidence intervals, or statistical significance tests.

---

### **Minor**
**Context length terminology:** The variable $K$ is defined as the number of ICL demonstrations (lines 353-354) but later termed "context length" (lines 1199-1200). These are fundamentally different. If $K$ represents context length, the experiments become questionable as $K \leq 20$ is too low for meaningful ICL.

---

### **Suggestions for Improvement**
- **More models:** I strongly recommend including at least one of Gemma or Mistral, as they reportedly lack the U-shaped pattern [2].

- **"Theorem" framing:** Presenting a simple bias-variance decomposition as a "theorem" is an overclaim. I suggest framing it as a "proposition" or "analytical result," especially given the linear attention and single-layer assumptions, though I acknowledge this is common practice. Theorem 1's assumptions (linear attention, single layer, i.i.d. demonstrations) diverge significantly from practice. Empirical validation of $\mathcal{O}(\frac{1}{K})$ rates in actual transformers remains incomplete.

- **Other geometric metrics:** Comparing TDNV to alternative metrics (e.g., nearest-class-center variance, silhouette score) would strengthen the methodological justification. Is TDNV uniquely informative or do other metrics show similar patterns?

---
**References**

[1] Skean et al. _Layer by Layer: Uncovering Hidden Representations in Language Models._ ICML 2025.

[2] Saglam et al. _Large Language Models Encode Semantics in Low-Dimensional Linear Subspaces._ arXiv preprint (2025).

[3] Valeriani et al. _The geometry of hidden representations of large transformer models._ NeurIPS 2023.

[4] Razzhigaev et al. _The Shape of Learning: Anisotropy and Intrinsic Dimensions in Transformer-Based Models._ Findings of EACL 2024.

**Questions:**

None

---

> ### Author Response · Authors · 2025-11-22
>
> > **Q1: The motivation and method appear very similar to Skean et al.'s work...one instance of task/information encoded in text.**
>
> **A1:** Thank you for pointing out this reference. However, we respectfully disagree that our work is similar as Skean et al.'s work [1]. The two works differ fundamentally in both motivation and methodology. While both works employ the term "information compression," the underlying definitions and the phenomena being measured are fundamentally different. Skean et al. focus on the geometry of **tokens** within a single sequence instance, utilizing metrics such as prompt entropy, curvature, and LiDAR to understand how token embeddings occupy space in a general sense. In contrast, our work investigates compression at the instance level. We define compression using our Task-Distance Normalized Variance (TDNV) metric, which quantifies how representations of distinct **ICL instances** belonging to the same task cluster together while remaining separable from those of different tasks. Our analysis therefore targets the inter-instance structure of task manifolds—specifically how a model aligns different inputs to a common task definition—rather than the intra-sequence geometry of individual token embeddings described in Skean et al. Therefore, the term "variance-based technical analysis" differs fundamentally depending on whether it is applied at the token level [1] or the instance level (ours). We have expanded our discussion of these differences in related works of Appendix.
>
> Furthermore, ICL is a uniquely interesting phenomenon that warrants dedicated investigation. Unlike general language modeling, ICL is an emergent ability that enables models to adapt to new tasks dynamically without changing parameters. Our primary motivation is to explain the specific mechanisms of this capability, which is not addressed by the referenced work. By investigating how LLMs extract and differentiate task information, we uncover a novel "Compression-Expression" dynamic: early layers progressively compress demonstrations into a compact task representation, which later layers then "express" to generate predictions. This specific layerwise dual-process is unique to the ICL setting and offers significant novelty beyond general representation analysis. We have updated our related works section to explicitly clarify these differences and better contextualize our contributions in the revision.
>
> Therefore, given these distinct methodological advances and novel insights into the internal mechanics of ICL, we believe our work makes a substantial and unique contribution to the field.
>
> > **Q2: Furthermore, recent work [2] demonstrates that...rendering the paper's findings questionable.**
>
> **A2:** We appreciate the reviewer's suggestions and the provided references. To further validate our findings, we extended our evaluation to include Gemma2-9B [3] and Mistral-7B [4] on symbolic ICL tasks using 15-shot, measuring layerwise TDNV, task vector accuracy, and early-exit accuracy. As illustrated in **Figure 19** of the revised manuscript, we observe a clear U-shaped TDNV pattern that aligns perfectly with our previous results. When combined with the five models originally analyzed (Deepseek-coder, GPT-J, Llama, Pythia, and Mamba), we have now demonstrated consistent behavior across seven distinct architectures. This universality confirms that the phenomenon is a fundamental mechanism of ICL, independent of specific model architectures or training methods. Furthermore, this robustness highlights the unique nature of the ICL mechanism identified in our study, distinguishing our contributions from prior work not conducted in the ICL setting.
>
> > **Q3: TDNV measures geometric properties (clustering, separation), not causal information flow.**
>
> **A3:** Thank you for your suggestion. While we did not explicitly use the term "causal intervention" in the initial text, the auxiliary metrics presented in Figure 1—Task Vector (TV) Accuracy and Early-Exit Accuracy—function precisely as causal interventions. By definition, a causal intervention involves manipulating an internal variable to observe its downstream effect on the output. In our Task Vector evaluation, we intervene by patching the hidden state extracted from the few-shot setting into a zero-shot pass; the resulting increase in accuracy confirms that the representations in the early "compression" layers causally encode the task definition required for correct generation. Similarly, Early-Exit Accuracy intervenes by forcing an immediate prediction from intermediate states, bypassing subsequent processing; the rise in accuracy here demonstrates that later layers are responsible for "expressing" the solution. Furthermore, as detailed in A4, we find a strong correlation between TDNV and these metrics, confirming that the geometric clustering we observe is predictive of causal model behavior. We will update in the revision to clarify the mechanism.

---

> ### Author Response · Authors · 2025-11-22
>
> > **Q4: Comparing TDNV to alternative metrics... other metrics show similar patterns?**
>
> **A4:** We selected Task-Distance Normalized Variance (TDNV) not as an ad-hoc measure, but as a theoretically grounded metric derived from the Neural Collapse framework, specifically designed to quantify the ratio of within-task compression to between-task discrimination. That being said, following your suggestion, we evaluated alternative metrics, including Nearest-Class-Center (NCC) variance and Silhouette Score, as shown in **Figure 20** of our revision. We find that NCC variance fails to capture the distinct "compression-expression" mechanism observed with TDNV. Furthermore, while the Silhouette Score assesses general clustering quality based on sample-to-sample distances, it does not explicitly model the geometry of class centroids (means). This distinction is critical for In-Context Learning (ICL), where the "task vector" is widely conceptualized as the mean representation of demonstrations. By focusing on the separability of these means relative to their variance, TDNV offers a more direct physical interpretation of the robustness of the extracted task representations.
>
> To quantitatively validate the superiority of TDNV, we evaluated how well each metric correlates with downstream task performance. Specifically, we measured the relationship between layerwise metrics (TDNV, NCC Variance, and Silhouette Score) and Task Vector (TV) accuracy using Distance Correlation (dCor). dCor is a robust statistical measure capable of detecting both linear and non-linear dependencies (where 0 indicates independence and 1 indicates strong dependency). As shown in the table below, TDNV exhibits a significantly stronger correlation with Task Vector accuracy compared to alternative metrics across both Mistral-7B and Gemma2-9B models. These consistently higher dCor values confirm that TDNV is the better predictor of model performance. Unlike generic clustering metrics, TDNV accurately reflects the 'quality' of the task information extracted by the model, directly linking the geometric structure of the representation to the model's actual ability to solve the task.
>
> | Model       | Nearest-Class-Center Variance | Silhouette | **TDNV** |
> |-------------|-------------------------------|------------|----------|
> | Mistral-7B  | 0.3436                        | 0.7297     | **0.7539** |
> | Gemma2-7B   | 0.4265                        | 0.7962     | **0.8558** |
>
> > **Q5: Results lack error bars, confidence intervals, or statistical significance tests.**
>
> **A5:** In Figure 1, we visualize the error bars of Task Vector accuracy and Early-Exit accuracy using shaded error bands. For our primary metric, TDNV, we conducted the experiments across three independent runs with different ICL samples. The resulting variance was negligible, demonstrating the high stability of the metric; consequently, we have omitted error bars for TDNV to maintain visual clarity in the plots.
>
> > **Q6: Context length terminology.**
>
> **A6:** Thanks for point it out. We change “context length” to “number of demonstrations” in the revision.
>
> > **Q7: Theorem" framing: Presenting a simple bias-variance decomposition as a "theorem" is an overclaim.**
>
> **A7**: We would like to point out that our theorem is not about bias-variance decomposition, but is to show how the decay of the bias and variance decays as K. To avoid the confusion, in the revision, we have removed the words “(Bias-variance decomposition with respect to K)”. We are happy to change theorem to proposition if the reviewer still think the later is better.
>
> [1] Skean et al. *Layer by Layer: Uncovering Hidden Representations in Language Models.* ICML 2025.
>
> [2] Saglam et al. *Large Language Models Encode Semantics in Low-Dimensional Linear Subspaces.* arXiv preprint (2025).
>
> [3] Team, Gemma, et al. "Gemma 2: Improving open language models at a practical size." *arXiv preprint arXiv:2408.00118* (2024).
>
> [4] Jiang, A. Q., Sablayrolles, A., Mensch, A., Bamford, C., Chaplot, D. S., de las Casas, D., ... & Sayed, W. E. (2023). Mistral 7B. *arXiv preprint arXiv:2310.06825*.

---

### Official Review · Reviewer_9tRv · 2025-11-01

**Soundness:** 3
**Presentation:** 3
**Contribution:** 3
**Rating:** 8
**Confidence:** 4

**Summary:**

This paper analyzes how Large Language Models (LLMs) perform in-context learning (ICL) by examining internal representations across layers. The authors discover a "Layerwise Compression-Expression" phenomenon: early layers progressively compress demonstration information into compact, discriminative task representations, while later layers express these representations using the query to generate predictions. Using Task-Distance Normalized Variance (TDNV) metrics, they show this pattern occurs consistently across diverse architectures, tasks, and modalities. The compression improves with model size and more demonstrations, explaining ICL performance gains. They provide theoretical analysis showing attention mechanisms reduce both bias and variance. The findings reveal how LLMs extract task information from demonstrations, offering insights for improving interpretability and robustness.

**Strengths:**

1. The "Layerwise Compression-Expression" phenomenon is a genuinely interesting discovery that provides new understanding of how ICL works internally. The observation that models split into distinct compression and expression phases is intuitive yet non-obvious.
2. Extensive experiments across multiple architectures (Transformers, Mamba), model scales, and task domains (symbolic, language understanding, multimodality). Shows the phenomenon emerges during training and isn't present in random models.
3. Provides mathematical analysis through bias-variance decomposition. Theorem showing how attention reduces both bias and variance with more demonstrations.
4. Clear figures that effectively illustrate the phenomenon

**Weaknesses:**

1. TDNV metric assumes tasks are well-separated and distinct, which may not hold for subtle task variations. The metric may not capture all aspects of task representation quality. Reliance on the last separator token as the task vector is somewhat arbitrary (though they do ablate this).
2. Experiments primarily focus on relatively simple, well-defined tasks.  Limited exploration of more complex, real-world ICL scenarios where task boundaries are unclear. Most experiments use relatively small models (except for a few with larger models).
3. While the paper describes WHAT happens, it provides limited insight into WHY this specific pattern emerges. Doesn't fully explain why compression must precede expression or why the transition occurs at specific layers.
4. Unclear how findings apply to tasks requiring complex reasoning or multi-step inference. Limited investigation of how the phenomenon varies with different prompt formats or instruction styles.

**Questions:**

1. How does the compression-expression pattern change with task complexity? For instance, in multi-hop reasoning tasks, do you observe multiple compression-expression cycles?
2.  How sensitive is TDNV to the choice of distance metric and number of samples? Have you validated that low TDNV actually correlates with better downstream performance across all your tasks?
3.  Could the U-shaped TDNV curve be explained by simpler factors like attention patterns becoming more focused toward the end, rather than a fundamental compression-expression mechanism?
4.  How do your findings relate to the "induction heads"? Do these heads primarily appear in compression or expression layers?

---

> ### Author Response · Authors · 2025-11-22
>
> > **Q1: How does the compression-expression pattern change with task complexity? For instance, in multi-hop reasoning tasks, do you observe multiple compression-expression cycles?**
>
> **A1:** To investigate how task complexity impacts layerwise dynamics, we constructed a group of 2-hop chain-of-thought reasoning tasks by chaining elementary operations such as *Next Letter*, *Previous Letter*, *To Uppercase*, and *Copy Letter*. Each instance follows a chain-of-thought structure formatted as “input” →”intermediate_step” →”final_output” (e.g. a→b→B, representing *Next Letter* followed by *To Uppercase*). Crucially, to capture the temporal evolution of task processing, we compute the layerwise TDNV at two distinct anchor points: the separator token preceding the intermediate output (Hop 1) and the separator preceding the final output (Hop 2). The results can be find in **Figure 21** in the revision.
>
> Our analysis reveals that the *Layerwise Compression-Expression* phenomenon is indeed recursive: we observe multiple distinct cycles corresponding to each reasoning hop. Specifically, the model exhibits a full compression-expression trajectory to generate the intermediate output (Hop 1) and subsequently initiates a new cycle to process that intermediate result for the final output (Hop 2). This suggests that LLMs decompose multi-hop ICL tasks into cascaded compression-expression operations rather than compressing the entire reasoning chain into a single representation.
>
> > **Q2: How sensitive is TDNV to the choice of distance metric and number of samples? Have you validated that low TDNV actually correlates with better downstream performance across all your tasks?**
>
> **A2:** Thank you for the comments. In the revision, we have included additional experiments to demonstrate the robustness of TDNV. Specifically, **Figure 22** in the revision demonstrates that the TDNV metric is highly robust to experimental design choices, maintaining a consistent U-shaped curve with negligible variance across different distance metrics (L1 v.s. L2), sample sizes (N = 100 or 200) and prompt formats(”→” v.s. “:”). This stability confirms that the observed Layerwise Compression-Expression is an intrinsic property of the model rather than an artifact of specific hyperparameters. Furthermore, we validate the functional significance of TDNV by showing that higher compression (lower TDNV) strongly predicts better downstream performance; specifically, we observe a distance correlation (dCor) of 0.8558 between TDNV and Task Vector accuracy in Gemma2-9B, confirming that the most compressed layers effectively encode the richest task information. Please refer to **Appendix C.3** for more details.
>
> > **Q3: Could the U-shaped TDNV curve be explained by simpler factors like attention patterns becoming more focused toward the end, rather than a fundamental compression-expression mechanism?**
>
> **A3:** We thank the reviewer for this insightful question. While attention mechanisms are indeed instrumental in extracting context, we think that the mere sharpening of attention patterns may not be able to fully explain the Layerwise Compression-Expression phenomenon. The TDNV metric specifically quantifies the geometric structure of *task separability*—how effectively the model clusters same-task instances while actively pushing different tasks apart—which is a distinct representational property that does not automatically result from focused attention weights alone. In addition, the persistence of the U-shaped TDNV curve in **Mamba** (State-Space Models), as shown in Figure 2, indicates that this phenomenon can be fully explained by attention patterns. Since Mamba relies on recurrent state transitions and completely lacks attention heads, it cannot exhibit "attention focusing" behavior. This universality demonstrates that the compression-expression dynamic is a fundamental mechanism of how deep sequence models distill and utilize in-context information, independent of the attention mechanism.
>
> We hope this clarification addresses your concern. If we have misinterpreted the question, we appreciate if the reviewer can elaborate and we would be more than happy to engage in further discussion.

---

> > ### Author Response · Authors · 2025-11-22
> >
> > > **Q4: How do your findings relate to the "induction heads"? Do these heads primarily appear in compression or expression layers?**
> >
> > **A4:** Thanks for the question. We view the "Layerwise Compression-Expression" framework and "Induction Heads" as complementary perspectives: our work offers a macroscopic geometric analysis of how representations evolve across layers, whereas Induction Heads provide a microscopic mechanistic explanation of specific attention circuits. However, there is a fundamental distinction regarding the task mechanism. Conceptually, an Induction Head is a specialized mechanism that searches for a previous context match and copies the subsequent token to the current position. In contrast, our ICL setting (e.g.,“a→b, b→c, c→ d, d→?”) requires the model to induce a latent task (such as "next letter") and apply it to generate a novel prediction, rather than simply retrieving and copying a token (like “b””c” or ”d”) that already appeared in the demonstrations. Therefore, our observation involves extracting an abstract task rule rather than the simple pattern-matching and copying behavior characteristic of standard Induction Heads.

---

### Official Review · Reviewer_X6rA · 2025-11-01

**Soundness:** 3
**Presentation:** 4
**Contribution:** 2
**Rating:** 4
**Confidence:** 3

**Summary:**

The paper provides a layerwise analysis of in-context learning (ICL), reporting Layerwise Compression-Expression (LCE). Early layers of transformer progressively compress the demonstration examples into a compact representation, and later layers of transformer express this information and incorporate this with the query to make the prediction. Using LCE, the authors analyze various ICL phenomenon, such as (1) larger model and more demo improves ICL (2) noisy demonstration hurts ICL. Finally authors propose task-vector contrastive fine-tuning method to improve ICL.

**Strengths:**

- The paper is overall well-written and structurally sound.
- The paper provides carefully designed metrics and experiments to analyze ICL.
- The fact that decoder-only LLM also acts as encoder-decoder model is interesting.

**Weaknesses:**

- The ICL tasks are synthetic and simple. Also models are small even compared to sLMs.
- Since this is tested on models solely trained for specific ICL tasks, LCE phenomenon might not be applicable to actual general-purpose LLM, which is of real interest.
- In the same manner "explains why larger models and longer contexts yield better performance" could be an over-statement.

**Questions:**

- Prior works have shown that there exists a positional bias for general-purpose LLM on ICL tasks, which can vary depending on the tasks and models. I wonder "perturbing demonstrations
that appear later in the sequence causes a larger performance drop" would be true for those models as well.
- Would it be possible to observe the encoder-decoder structure in pretrained LLM as well?
- The proposed task-vector contrastive fine-tuning is solely proposed for ICL fine-tuning. What would be its practical application?

---

> ### Author Response · Authors · 2025-11-22
>
> > **Q1: The ICL tasks are synthetic and simple. Also models are small even compared to sLMs.**
>
> **A1:** Thank you for the comments. While we utilize synthetic tasks to facilitate a controlled, systematic geometric analysis of ICL mechanics, we also validated our findings on more complex domains, including natural language understanding with long-context sentences and multimodal tasks (Section 4.1), which offer richer settings than prior studies [1]. Additionally, we include multi-hop reasoning tasks in **Appendix C.4** of the revision. Regarding model size, we employed 7B-parameter models (e.g., DeepSeek-Coder) as they represent the standard scale for open-source interpretability research.
>
> > **Q2: Since this is tested on models solely trained for specific ICL tasks...Would it be possible to observe the encoder-decoder structure in pretrained LLM as well?**
>
> **A2:** We would like to clarify this crucial **misunderstanding** regarding our experimental setup. Our primary discovery—the Compression-Expression phenomenon—was observed and analyzed in **general-purpose, pretrained** LLMs (including Llama-3, DeepSeek-Coder, Pythia, and Mamba) that were **not** trained for specific ICL tasks. This is precisely the significance of our finding: standard decoder-only architectures spontaneously exhibit encoder-decoder-like behavior during in-context learning without any special supervision. Consequently, our conclusions regarding the effects of model size and context length are derived directly from these general-purpose models and are not overstatements based on specialized training. We only utilized task-specific fine-tuning in Section 6 (on GPT-2) to validate a specific application (task-vector contrastive fine-tuning), which is distinct from the general analysis establishing the core phenomenon.
>
> > **Q3: I wonder "perturbing demonstrations that appear later in the sequence causes a larger performance drop" would be true for those models as well.**
>
> **A3:**  We confirm that our analysis was conducted entirely on general-purpose pretrained LLMs performing ICL tasks. Our results align with the positional biases noted in prior work, specifically confirming that perturbing demonstrations appearing later in the sequence causes a significantly larger performance drop in these models. Crucially, our framework goes beyond merely observing this "positional bias" by offering a geometric explanation for it. As illustrated in Figure 9 (Section 4.3), we observe a strong correlation between these performance drops and higher TDNV values. It is further explained by the fine-grained token-level grid-TDNV analysis in Figure 25 of appendix. This indicates that the positional bias arises due to less compact task vectors.
>
> > **Q4: The proposed task-vector contrastive fine-tuning is solely proposed for ICL fine-tuning. What would be its practical application?**
>
> **A4:** Our method serves as a novel training algorithm that explicitly encourages task-vector formation. Through task-vector contrastive fine-tuning, the model yields stronger and more separable task vectors, which leads to several meaningful practical benefits. First, task vector improves efficiency by enabling models to perform complex tasks with fewer or even zero demonstrations, effectively condensing long prompts into compact internal representations [1]. Second, it allows for better control over model behavior; because tasks are encoded as specific vectors, we can potentially add, remove, or combine them to steer the model’s output without needing to craft new prompts [2,3]. Finally, task vector makes the model's decision-making process more transparent, transforming the abstract mechanism of In-Context Learning into concrete geometric patterns that are easier for researchers to analyze and understand [4].
>
> [1] Hendel, Roee, Mor Geva, and Amir Globerson. "In-context learning creates task vectors." *arXiv preprint arXiv:2310.15916* (2023).
>
> [2] Todd, Eric, et al. "Function vectors in large language models." *arXiv preprint arXiv:2310.15213* (2023).
>
> [3] Liu, Sheng, et al. "In-context vectors: Making in context learning more effective and controllable through latent space steering." *arXiv preprint arXiv:2311.06668* (2023).
>
> [4] Zou, Andy, et al. "Representation engineering: A top-down approach to ai transparency." *arXiv preprint arXiv:2310.01405* (2023).

---

### Official Review · Reviewer_gCAs · 2025-11-01

**Soundness:** 3
**Presentation:** 3
**Contribution:** 3
**Rating:** 6
**Confidence:** 3

**Summary:**

This paper investigates the layerwise dynamics of in-context learning (ICL) in large language models. It introduces a metric called Task-Distance Normalized Variance (TDNV) to characterize how task information is compressed in early layers and expressed in later layers. The study includes experiments across multiple model architectures and task types, and analyzes how factors such as model size and demonstration count affect this pattern. A bias–variance decomposition of task vectors is also presented, along with applications for identifying optimal layers and improving task representations.

**Strengths:**

1. The paper proposes TDNV as a new metric for analyzing in-context learning (ICL), and presents a clear two-phase compression–expression pattern in layerwise representations. The experimental setup is thorough and supports the empirical observations across different model sizes and settings.

2. The authors provide a bias–variance decomposition of task vectors, offering a structured interpretation of how representation quality evolves with the number of demonstrations. This analysis contributes to a more systematic understanding of ICL dynamics.

3. The work applies its findings to practical objectives by identifying the optimal layer for extracting task vectors using TDNV and introducing a contrastive fine-tuning strategy that improves task-vector performance. These applications demonstrate the potential utility of the proposed analyses.

**Weaknesses:**

1. The paper adopts TDNV as a central metric but provides limited theoretical support for why this specific ratio captures task compression or predicts ICL performance. The analysis is primarily empirical, and while correlations such as the alignment of minimum TDNV with peak task-vector accuracy are demonstrated, the underlying mechanisms remain unexplained.

2. The experimental evaluation primarily focuses on symbolic tasks such as letter transformations and simple list operations. While the paper includes additional experiments on natural language and multimodal settings, these are conducted on relatively small synthetic datasets.

**Questions:**

1. The paper exclusively uses TDNV to demonstrate the compression-expression phenomenon, without examining whether the pattern depends on this specific metric. Have the authors investigated whether the observed trend holds under alternative metrics?

2. The paper's evaluation focuses on symbolic tasks and small synthetic datasets, leaving the generalizability to complex reasoning tasks unclear. Have the authors evaluated whether the compression-expression phenomenon
holds on standard ICL benchmarks such as GSM8K, MMLU, or BBH?

---

> ### Author Response · Authors · 2025-11-22
>
> > **Q1: Provides limited theoretical support for why this specific ratio captures task compression or predicts ICL performance.**
>
> **A1:** We selected Task-Distance Normalized Variance (TDNV) as a theoretically grounded metric derived from the Neural Collapse framework. It specifically quantifies the ratio of within-task variance to between-task separation. Conceptually, when representations of different instances from the same task cluster tightly together while remaining distinct from other tasks, it implies the model has successfully abstracted the common task information and discarded instance-specific noise. Therefore, a low TDNV score directly reflects the successful compression of task-relevant information into the representation space.
>
> To empirically validate that this compression predicts performance, we measured the relationship between layerwise TDNV and downstream Task Vector (TV) accuracy using Distance Correlation (dCor), a robust metric for detecting non-linear dependencies. Our results on Mistral-7B and Gemma2-9B show that TDNV exhibits a significantly stronger correlation with TV accuracy than alternative metrics like Nearest-Class-Center variance or Silhouette Score. This confirms that the geometric structure captured by TDNV is a reliable and faithful predictor of the model's actual capability to execute the task.
>
> | Model       | Nearest-Class-Center Variance | Silhouette | **TDNV** |
> |-------------|-------------------------------|------------|----------|
> | Mistral-7B  | 0.3436                        | 0.7297     | **0.7539** |
> | Gemma2-7B   | 0.4265                        | 0.7962     | **0.8558** |
>
> More comprehensive theoretical analysis would be an important direction to pursue in future work.
>
> > **Q2: Have the authors investigated whether the observed trend holds under alternative metrics?**
>
> **A2:** Yes, we have extensively investigated alternative metrics to validate the robustness of our findings. As detailed in **Appendix C.3**, we compared our TDNV metric against Nearest-Class-Center (NCC) variance and the Silhouette Score using Mistral-7B and Gemma2-9B models. While these general clustering metrics provide some geometric insights, our analysis using Distance Correlation (dCor) demonstrates that TDNV exhibits a significantly stronger correlation with downstream Task Vector accuracy. This confirms that the "U-shaped" Compression-Expression trend is best captured by TDNV, as it uniquely accounts for the separability of task means relative to their variance—a geometric property central to the mechanics of In-Context Learning.
>
> > **Q3: The paper's evaluation focuses on symbolic tasks and small synthetic datasets, leaving the generalizability to complex reasoning tasks unclear.**
>
> **A3:** We utilized synthetic tasks to facilitate a controlled and systematic geometric analysis of ICL mechanics, a standard practice in foundational ICL research [1]. However, we have already validated our findings on more complex domains, including natural language understanding with long-context sentences and multimodal tasks (Section 4.1). These settings are significantly richer than simple symbolic mappings and comparable to the semantic classification tasks analyzed in recent literature [2].
>
> Regarding standard benchmarks like GSM8K or MMLU, we clarify that our study specifically focuses on **task distinction**—scenarios where a single query possesses multiple latent attributes, and the model must infer the specific mapping (e.g. A→color, or A→shape) solely from demonstrations. Standard benchmarks typically involve fixed reasoning paths (e.g., "solve this math problem") where the task definition is constant rather than context-dependent. Because existing datasets lack this multi-attribute structure required to measure task separability, we constructed our own rigorous evaluation sets. Nevertheless, to address the reviewer’s interest in complex reasoning, we have extended our evaluation to include **multi-hop reasoning tasks** in Figure 21, where we observe consistent support for the Compression-Expression phenomenon.
>
> [1]  Hendel, Roee, Mor Geva, and Amir Globerson. "In-context learning creates task vectors." *arXiv preprint arXiv:2310.15916* (2023).
>
> [2] Wei, Jerry, et al. "Larger language models do in-context learning differently." arXiv preprint arXiv:2303.03846 (2023)

---

### Author Response · Authors · 2025-12-02
**General Response**

We sincerely regret the recent incident and  appreciation for the extra effort it may have required from the AC. We would like to thank the reviewers for providing constructive feedback to our work.  Below, we provide a summary of our discussion.

Reviewer **nSwE** raised concerns about similarity to prior work; we clarified in the revision that, unlike [1] which performs token-level analysis, our approach is sample-level, and we added new experiments on Gemma and Mistral to further demonstrate the consistency of our findings and their differences from [2]. Reviewer **X6rA** misunderstood our setting; we emphasized that our phenomenon is primarily observed in pretrained LLMs. Reviewers **gCAs** and **nSwE** mentioned alternative metrics; in the revision, we added comparisons with nearest-class-center variance and silhouette score, showing that TDNV best correlates with downstream performance. Reviewer **9tRv** requested multi-hop reasoning evaluation; we incorporated new 2-hop chain-of-thought tasks and observed that the compression–expression cycles.

Unfortunately, none of the reviewers have responded before the rollback. Nevertheless, **we believe we have addressed the main concerns raised in the reviews.** Overall we found the discussion constructive and thoughtful. We sincerely thank the reviewers and AC for the thoughtful engagement.

---

### Meta-Review · Area_Chair_wpd7 · 2026-01-03

**Summary:**

This paper re-investigates the layer-wise functionality and propagation of ICL demonstrations via a novel metric, Task-Distance Normalized Variance (TDNV). Through this metric, the authors uncover a Layerwise Compression-Expression phenomenon, in which early layers compress information from demonstrations to task information while later layers express the information to the query and form the final answer. This phenomenon demonstrates consistency across various LMs and tasks, and offers implications for ICL performance. A bias-variance decomposition further provides theoretical explanations.

Overall, this paper offers an interesting and comprehensive interpretation of the ICL mechanism through the lens of geometric properties. The identified phenomenon is intriguing and intuitive, well supported by extensive empirical validation across models, tasks and interventions. The layerwise compression-expression phenomenon also offers practical applications for enhancing ICL, such as an effective method to locate the optimal task-vector layer and task-vector contrastive finetuning.

Beyond those, the reviewers point out the following limitations which are worth considering and refining upon:
- The main limitation lies in the applicability of the findings to more realistic tasks. The authors mainly conduct experiments on symbolic and synthetic tasks which are quite different from ICL applications in real scenarios (such as MMLU, math reasoning, etc). Applicability towards the more general and advanced domains is unknown.
- The applicability in larger models is unknown (LMs with size larger than 7B).
- TDNV may rely on the tasks being tested. The task distance metric is sensitive to how different the tasks are being selected, and hence may only work in certain scenarios.

I do like the phenomenon and analysis of this paper, but given the above limitations, I would expect the authors refining this work to provide an even broader impact.

**Reviewer Concerns:**

Concerns being addressed:
- Sensitivity of the phenomenon to different metrics: Besides TDNV, the authors conduct additional experiments on alternative metrics such as Nearest-Class-Center Variance and Silhouette score.
- Correlation between TDNV and task accuracy: The authors uses Distance Correlation (dCor) to measure the correlation which shows higher values compared to alternative metrics.
- Misunderstanding of whether the LMs chosen are general-purpose pretrained or not.
- Investigation of how the phenomenon varies with different prompt formats, instruction styles and number of samples (and multi-hop scenarios).
- Additional experiments testing whether the phenomenon holds for other LMs like Mistral and Gemma.
- Clarifications on whether the phenomenon is causal vs correlational and statistical testing.

Outstanding concerns:
- The main limitation lies in the applicability of the findings to more realistic tasks. The authors mainly conduct experiments on symbolic and synthetic tasks which are quite different from ICL applications in real scenarios (such as MMLU, math reasoning, etc). Applicability towards the more general and advanced domains is unknown.
- The applicability in larger models is unknown (LMs with size larger than 7B).
- TDNV metric assumes tasks are well-separated and distinct, which may not hold for subtle task variations. The metric may not capture all aspects of task representation quality.

**Reviewer Scores:**

The four reviewers provide quite diverse scores: 2, 4, 6, 8.

To me, I feel score 8 does not perfectly align with the corresponding review (which gives me a feeling that there are still some valid concerns to be addressed) and I don't see the authors addressing all of them. Therefore, I expect this score to be decreased to 6.

I also expect score 2 to be increased to 4 given some of the questions are being clarified, with additional experiments.

---

### Decision · Program_Chairs · 2026-01-26

Reject